# `GradientStabilizer`: Fix the Norm, Not the Gradient

Tianjin Huang [✉ 1 2]  Zhangyang Wang [3]  Haotian Hu [4]  Zhenyu Zhang [3]  Gaojie Jin [1]  Xiang Li [5]
Li Shen [6]  Jiaxing Shang [7]  Tianlong Chen [8]  Ke Li [1]  Lu Liu [1]  Qingsong Wen [9]  Shiwei Liu [10 11 12]

## Abstract

Training instability in modern deep learning systems is frequently triggered by rare but extreme gradient-norm spikes, which can induce oversized parameter updates, corrupt optimizer state, and lead to slow recovery or divergence. Widely used safeguards such as gradient clipping mitigate these failures but require threshold tuning and indiscriminately truncate large updates. We propose `GradientStabilizer`, a lightweight, drop-in gradient transform that *preserves the instantaneous gradient direction* while replacing the update magnitude with a statistically stabilized estimate derived from running gradient-norm statistics. We prove that the resulting stabilized magnitude is uniformly bounded on spike steps, independent of the spike size, and show how this boundedness controls optimizer state evolution in adaptive methods. Across LLM pre-training (FP16), quantization-aware pre-training (FP4), ImageNet classification, reinforcement learning, and time-series forecasting, `GradientStabilizer` consistently improves training stability, widens stable learning-rate regions, and reduces divergence relative to clipping-based baselines, even substantially reducing Adam's sensitivity to weight-decay strength. The Code is available at https://github.com/TianjinYellow/

[1]Department of Computer Science, University of Exeter [2]Department of Mathematics and Computer Science, Eindhoven University of Technology [3]Department of Electrical and Computer Engineering, University of Texas at Austin [4]School of the Gifted Young, University of Science and Technology of China [5]Department of Computer Science, University of Reading [6]School of Cyber Science and Technology, Sun Yat-sen University [7]College of Computer Science, Chongqing University [8]Department of Computer Science, The University of North Carolina at Chapel Hill [9]Squirrel Ai Learning [10]ELLIS Institute Tubingen [11]Max Planck Institute for Intelligent Systems [12]Tübingen AI Center, Tübingen, Germany. Correspondence to: Tianjin Huang <t.huang2@exeter.ac.uk>.

*Proceedings of the 43rd International Conference on Machine Learning*, Seoul, South Korea. PMLR 306, 2026. Copyright 2026 by the author(s).

`GradientStabilizer.git`.

## 1. Introduction

The optimization of deep neural networks has advanced rapidly over the past decade, driven by stochastic gradient descent (SGD) and adaptive variants such as Adam and its extensions (Kingma, 2014; Shazeer & Stern, 2018; Gupta et al., 2018; Loshchilov & Hutter, 2017). Despite these successes, *training instability* remains a persistent challenge in modern large-scale regimes. Instabilities are frequently observed in large language model (LLM) pre-training (Chowdhery et al., 2023; Liu et al., 2025; Takase et al., 2023), reinforcement learning with verifiable rewards (Yu et al., 2025; Zeng et al., 2025; He et al., 2025), and quantization-aware training (Wortsman et al., 2023; Zhang, 2025). In these settings, rare but extreme *gradient-norm spikes* can induce oversized parameter updates, distort optimizer state, and occasionally trigger catastrophic divergence.

A common safeguard against such failures is gradient clipping, which caps the norm (or coordinates) of the update to prevent excessive parameter changes (Pascanu et al., 2013; Brock et al., 2021; Kumar et al., 2025). While effective in practice, clipping operates as an extrinsic post-processing rule that enforces instantaneous constraints via fixed thresholds. As a result, it requires careful tuning and may either intervene too late to prevent instability or unnecessarily suppress informative updates during otherwise stable phases of training. More adaptive variants, such as ZClip and adaptive gradient clipping (Kumar et al., 2025; Wang et al., 2025; Brock et al., 2021), partially address threshold sensitivity but remain fundamentally reactive mechanisms that truncate gradients once constraints are violated.

This work proposes `GradientStabilizer`, an intrinsic stabilization mechanism that addresses gradient spikes and explosions by *structurally decoupling the update direction from its magnitude* (As shown in Figure 1). The core principle is simple: while the *direction* of the gradient often provides reliable descent information, its *instantaneous norm* can be highly volatile and dominated by noise or rare outliers. `GradientStabilizer` preserves the instantaneous gradient direction while replacing its magnitude with

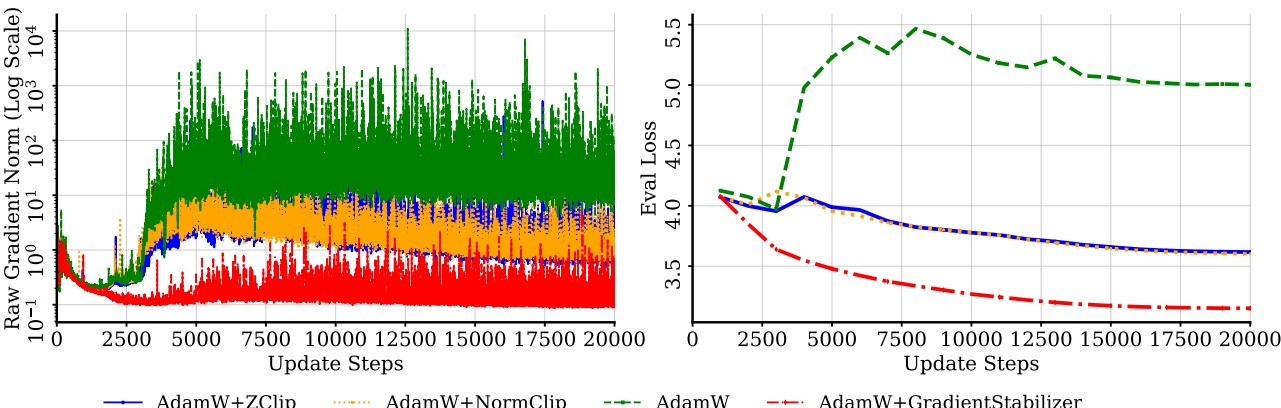

*Figure 1.* **Raw gradient norm and evaluation loss across update steps.** AdamW+`GradientStabilizer` suppresses gradient norm explosion while achieving the lowest evaluation loss compared with AdamW, AdamW+ZClip, and AdamW+NormClip. Experiments are conducted on LLaMA-130M with 2.2B Tokens using a learning rate of $3 \times 10^{-3}$.

a statistically stabilized estimate computed from running averages of gradient norms. This yields smooth, threshold-free control of update magnitudes without truncating directions or other manual intervention.

We then provide a theoretical analysis of `Gradient Stabilizer` in both stationary and spike-driven regimes. In stationary settings, we characterize the population target of the stabilized magnitude as a mean-to-RMS ratio that decreases with the coefficient of variation of gradient norms, explaining its variance-dampening behavior. More importantly, under a simple spike event model, we show that the stabilized update magnitude is *uniformly bounded on spike steps*, independent of the raw spike size. This property ensures that arbitrarily large gradient spikes cannot produce arbitrarily large parameter updates once passed through `GradientStabilizer`. When combined with adaptive optimizers such as Adam, this bounded effective-gradient property suffices to control the internal moment states of Adam/AMSGrad and to bound each per-coordinate update, which are key technical conditions assumed by standard nonconvex convergence analyses.

Our contributions are summarized as follows:

* **Method.** We introduce `GradientStabilizer`, a lightweight, drop-in gradient transform that preserves the update direction while adaptively stabilizing the magnitude using running statistics of gradient norms, providing a threshold-free alternative to gradient clipping.

* **Theoretical characterization.** We analyze `Gradient Stabilizer` in both stationary and spike regimes, establishing a variance-dampening interpretation in stationary settings and a spike-dampening bound that guarantees uniformly bounded update magnitudes on arbitrarily large gradient spikes.

* **Optimization implications.** We show that the intrinsic bounded stabilized gradient property induced by `GradientStabilizer` suffices to control the internal moment states of Adam/AMSGrad and to bound each per-coordinate update, independent of the magnitude of raw gradient spikes.

* **Empirical evaluation.** Across a wide spectrum of tasks, `GradientStabilizer` improves training stability, broadens the stable learning-rate region, and reduces divergence compared to clipping-based baselines. Additionally, we observe that gradient clipping exacerbates Adam's sensitivity to weight-decay strength, whereas `GradientStabilizer` substantially reduces this sensitivity across tasks.

## 2. `GradientStabilizer`

In this section, we propose `GradientStabilizer`, a lightweight and optimizer-agnostic gradient transformation for stabilizing training under gradient-norm spikes. `GradientStabilizer` replaces the instantaneous magnitude of the gradient with a statistically stabilized estimate derived from the history of the optimization trajectory. We define the update direction $d_t$ as the unit vector of the current gradient:

$$d_t = \frac{g_t}{\|g_t\|_2}$$

To determine the step magnitude, we track the first and second moments of the gradient norm using Exponential Moving Averages (EMA). Let $R_t = \|g_t\|_2$. We update the moment estimates $m_t$ and $v_t$ as follows:

$$m_t^R = \gamma_1 m_{t-1}^R + (1 - \gamma_1) R_t$$

$$v_t^R = \gamma_2 v_{t-1}^R + (1 - \gamma_2) R_t^2$$

**Algorithm 1** `GradientStabilizer`

---

1: **Input:** initial parameters $\theta_0$; base optimizer $\mathcal{A}$ with state $\phi_0$; loss function $\ell_t$ ; learning rates $\{\eta_t\}_{t=1}^T$; EMA decays $\gamma_1, \gamma_2 \in [0, 1)$
2: **Output:** final parameters $\theta_T$
3: $m_0^R \leftarrow 0, \ v_0^R \leftarrow 0$   ▷ EMA states for gradient norms
4: **for** $t = 1$ **to** $T$ **do**
5:   $g_t \leftarrow \nabla_\theta \ell_t(\theta_{t-1})$   ▷ stochastic gradient
6:   $R_t \leftarrow \|\texttt{Flatten}(g_t)\|_2$
7:   $m_t^R \leftarrow \gamma_1 m_{t-1}^R + (1 - \gamma_1) R_t$
8:   $v_t^R \leftarrow \gamma_2 v_{t-1}^R + (1 - \gamma_2) R_t^2$   ▷ EMA moments
9:   $d_t \leftarrow g_t / R_t$   ▷ unit direction
10:   $\rho_t \leftarrow m_t^R / \sqrt{v_t^R}$   ▷ stabilized magnitude
11:   $\tilde{g}_t \leftarrow \rho_t \cdot d_t$
12:   $(\theta_t, \phi_t) \leftarrow \texttt{Update}_{\mathcal{A}}(\theta_{t-1}, \tilde{g}_t, \eta_t, \phi_{t-1})$
13: **end for**

---

where $\gamma_1, \gamma_2 \in [0, 1)$ are decay rates controlling the effective memory length. Using these moments, we compute the stablized magnitude $\rho_t$:

$$\rho_t = \frac{m_t^R}{\sqrt{v_t^R}}$$

consider a generic stochastic optimization algorithm $\mathcal{A}$ (e.g. Adam (Kingma, 2014),AdamW (Loshchilov & Hutter, 2017),SPAM (Huang et al., 2025) and etc), the final parameter update is constructed by scaling the unit direction $d_t$ with this stabilized mangnitude $\rho_t$ and the learning rate $\eta_t$:

$$\theta_{t+1} = \texttt{Update}_{\mathcal{A}}(\theta_t, \tilde{g}_t, \eta_t, \phi_t), \quad \tilde{g}_t = d_t \cdot \rho_t$$

where $\phi_t$ represents the internal state of the optimizer (e.g., momentum buffers). `GradientStabilizer` is a lightweight, drop-in gradient transformation and can be integrated into standard training pipelines with minimal overhead. Algorithm 1 provides the pseudocode.

## 3. Theoretical Justification

In this section, we analyze the stability properties of `GradientStabilizer`. We characterize its behavior in stationary and spike-driven regimes, and show that it induces uniformly bounded effective gradients and optimizer moment states. These results establish key stability prerequisites that are required by existing convergence analyses for nonconvex optimization with adaptive methods.

### 3.1. Stability Properties

We first characterize the behavior of the stabilized magnitude $\rho_t$ under statistical assumptions on the gradient norm.

**Definition 3.1** (Spike Event). Fix $\kappa \gg 1$. Let $R_t$ denote the raw gradient norm at iteration $t$, and let $m_{t-1}^{(R)}$ denote the first moment estimator of the historical norms $\{R_s\}_{s=1}^{t-1}$,

computed via an exponential moving average. The *spike event* at iteration $t$ is defined as $\mathcal{S}_t := \{ R_t \geq \kappa \, m_{t-1}^{(R)} \}$.

**Lemma 3.2** (Variance Dampening). *Let $R \geq 0$ be a random variable denoting the (approximately stationary) gradient norm, with $\mu = \mathbb{E}[R]$, $\sigma^2 = Var(R)$ and $\nu = \mathbb{E}[R^2]$. Define the target population ratio*

$$\rho_\star = \frac{\mathbb{E}[R]}{\sqrt{\mathbb{E}[R^2]}} = \frac{\mu}{\sqrt{\nu}}.$$

*Let $c_v^2 = \sigma^2/\mu^2$ be the squared coefficient of variation. Then*

$$\rho_\star = \frac{1}{\sqrt{1 + c_v^2}} \leq 1. \tag{1}$$

*Proof.* See Appendix A.1.   □

*Remark* 3.3. Lemma 3.2 provides a population-level characterization of the stability ratio: the ratio $\rho_\star = \mathbb{E}[R]/\sqrt{\mathbb{E}[R^2]}$ decreases monotonically with the variability of $R$ (via $c_v^2$), capturing an intrinsic *variance-dampening* effect. In our method, the stabilized magnitude $\rho_t = m_t^R/\sqrt{v_t^R}$ is computed from fixed-decay EMA statistics $m_t^R \approx \mathbb{E}[R]$ and $v_t^R \approx \mathbb{E}[R^2]$ over an effective window of length $\asymp (1 - \gamma_2)^{-1}$. Thus, under approximate stationarity over this window, $\rho_t$ *tracks* the target ratio $\rho_\star$ (up to finite-window estimation error), inheriting the same qualitative behavior.**(I) High-variance regime:** In the presence of noise or gradient spikes, $\rho_t$ diminishes, naturally contracting the update step to preserve stability. **(II) Low-variance regime (i.e., $c_v \to 0$):** As the variance vanishes, $\rho_t \to 1$. In this phase, the algorithm utilizes the full learning rate $\eta$, recovering the dynamics of Normalized Gradient Descent.

**Lemma 3.4** (Uniform Spike-Step Upper Bound). *Let $\{g_t\}_{t \geq 1}$ be any stochastic gradient sequence with norms $R_t = \|g_t\|_2$ and Let $m_{t-1}^{(R)}$ be the norm for the first moment $m_{t-}$ at step $t - 1$. Assume that gradient spikes occur with satisfying the condition $R_t \geq \kappa \, m_{t-1}^{(R)}$ for a threshold $\kappa \gg 1$. Fix decay rates $\gamma_1, \gamma_2 \in [0, 1)$. For any historical state $(m_{t-1}, v_{t-1})$ consistent with the EMA recursions, the stablized gradient norm $\|\tilde{g}_t\|_2$ is bounded by:*

$$\|\tilde{g}_t\|_2 \leq \rho_t \leq \frac{1 - \gamma_1}{\sqrt{1 - \gamma_2}} + \frac{\gamma_1}{\kappa\sqrt{1 - \gamma_2}} \tag{2}$$

*where $\|\tilde{g}_t\|_2 = \rho_t$ whenever $R_t > 0$, and the inequality is strict only in the degenerate case $R_t = 0$.*

*Proof.* See Appendix A.2.   □

*Remark* 3.5. Lemma 3.4 provides a spike-invariant upper bound on the stabilized gradient norm: even if the raw

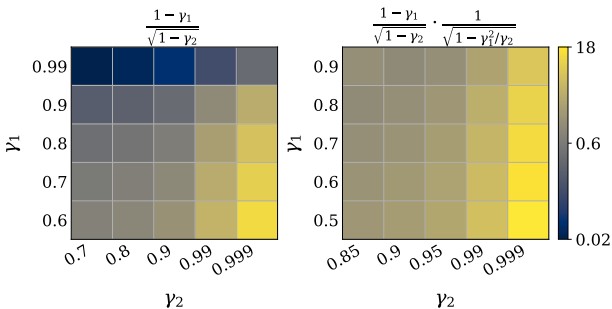

*Figure 2.* **Heatmaps of Coefficient-dependent bound factors in Lemma 3.4 and Lemma 3.6.** Heatmaps of $\frac{1-\gamma_1}{\sqrt{1-\gamma_2}}$ (left) and $\frac{1}{\sqrt{1-\gamma_1^2/\gamma_2}}$ (right) over a $(\gamma_1, \gamma_2)$ grid. The right panel is restricted to the feasible region $\gamma_1^2 < \gamma_2$, under which the bound in Lemma 3.6 is well-defined.

gradient norm $R_t$ becomes arbitrarily large on spike steps where $\kappa$ can be very large and reach up to $1000\times$ in practice (Huang et al., 2025), the stabilized gradient norm $\rho_t$ cannot blow up, since when $\kappa$ is large, the second term is negligible and the effective ceiling is well-approximated by $\rho_t \lesssim (1-\gamma_1)/\sqrt{1-\gamma_2}$. The heatmap in Figure 2 (Left) empirically validates that this upper bound remains within a controlled range across a wide spectrum of $\gamma_1$ and $\gamma_2$ values.

**Lemma 3.6** (Uniform Bound on Stabilized Gradient). *Assume $m_0^{(R)} = v_0^{(R)} = 0$ and $\gamma_1, \gamma_2 \in [0,1)$ satisfy $\gamma_1^2 < \gamma_2$. Then for all $t \geq 1$,*

$$\|\tilde{g}_t\|_2 \leq \rho_t \leq \bar{\rho} := \frac{1-\gamma_1}{\sqrt{1-\gamma_2}} \cdot \frac{1}{\sqrt{1-\gamma_1^2/\gamma_2}}. \quad (3)$$

*where $\|\tilde{g}_t\|_2 = \rho_t$ whenever $R_t > 0$, and the inequality is strict only in the degenerate case $R_t = 0$.*

*Proof.* See Appendix A.3. $\square$

*Remark* 3.7. Lemma 3.6 establishes a *time-uniform* bound on the stabilized gradient. Under the condition $\gamma_1^2 < \gamma_2$, the stabilized gradient norm $\|\tilde{g}_t\|_2$ is bounded by a closed-form constant $\bar{\rho}$ that depends only on the EMA decay rates $(\gamma_1, \gamma_2)$. In particular, this bound holds for all $t$ and does not depend on the instantaneous raw norm $\|g_t\|_2$, ruling out the divergence of stabilized gradient even when the raw gradients are extremely noisy or exhibit rare spikes. Figure 2 (Right) visualizes $\bar{\rho}$ across a spectrum of $(\gamma_1, \gamma_2)$, demonstrating that the upper bound remains moderate for typical choices of decay rates.

### 3.2. Implications for Optimizer Stability

**Proposition 3.8** (Coordinatewise Bound from Lemma 3.6). *Under the conditions of Lemma 3.6, for all $t \geq 1$,*

$$\|\tilde{g}_t\|_\infty \leq \|\tilde{g}_t\|_2 \leq \bar{\rho}, \text{ and hence } |\tilde{g}_{t,i}| \leq \bar{\rho} \ \forall i. \quad (4)$$

*Proof.* See Appendix A.4 $\square$

**Theorem 3.9** (Bounded Adam/AMSGrad Moment States under `GradientStabilizer`). *Let $\{\tilde{g}_t\}_{t \geq 1}$ be the stabilized gradients produced by `GradientStabilizer`. Under the conditions of Proposition 3.8 and $\beta_1, \beta_2 \in [0,1)$ and initialize $m_0 = 0$, $v_0 = 0$, and $\hat{v}_0 = 0$. Consider Adam (Kingma, 2014)/AMSGrad (Reddi et al., 2019) moment recursions*

$$m_t = \beta_1 m_{t-1} + (1-\beta_1)\tilde{g}_t, \quad (5)$$

$$v_t = \beta_2 v_{t-1} + (1-\beta_2)(\tilde{g}_t)^{\odot 2}, \quad (6)$$

$$\hat{v}_t = \max\{\hat{v}_{t-1}, v_t\} \quad (\text{AMSGrad only}), \quad (7)$$

*where $(\cdot)^{\odot 2}$ and $\max(\cdot, \cdot)$ are applied elementwise. Then, for all $t \geq 1$,*

$$\|m_t\|_\infty \leq \bar{\rho}(1-\beta_1^t); \|v_t\|_\infty, \|\hat{v}_t\|_\infty \leq \bar{\rho}^2(1-\beta_2^t). \quad (8)$$

*Proof.* See Appendix A.5. $\square$

*Remark* 3.10. Theorem 3.9 indicates that if Adam/AMSGrad is driven by the stabilized gradients $\{\tilde{g}_t\}$, the internal moment states cannot blow up: the first-moment estimate $m_t$ and the second-moment accumulators $v_t$ (and $\hat{v}_t$ for AMSGrad) remain uniformly bounded for all $t$, independent of the magnitude of the *raw* gradient spikes that may have occurred prior to `GradientStabilizer`. These bounds preclude unbounded moment growth and make the update mapping well-defined, providing a basic stability prerequisite for Adam-type methods. Importantly, we do not claim convergence by itself; rather, we isolate and guarantee a stability condition that is typically assumed (but rarely verified) in convergence analyses of adaptive optimizers.

**Corollary 3.11** (Bounded Per-step Parameter Change for SGD under `GradientStabilizer`). *Consider SGD driven by the stabilized gradients produced by `GradientStabilizer`: $\theta_{t+1} = \theta_t - \eta_t \tilde{g}_t$. Under the conditions of Proposition 3.8, for all $t \geq 1$,*

$$\|\theta_{t+1} - \theta_t\|_\infty \leq \eta_t \bar{\rho}. \quad (9)$$

*Moreover, on spike event $\mathcal{S}_t$, i.e., those satisfying $R_t \geq \kappa m_{t-1}^{(R)}$ for some $\kappa \gg 1$,*

$$\|\theta_{t+1} - \theta_t\|_\infty \leq \eta_t \left( \frac{1-\gamma_1}{\sqrt{1-\gamma_2}} + \frac{\gamma_1}{\kappa\sqrt{1-\gamma_2}} \right). \quad (10)$$

*Proof.* See Appendix A.7. $\square$

*Remark* 3.12. Corollary 3.11 shows that, under `GradientStabilizer`, the worst-case coordinate update is controlled by the intrinsic bound $\bar{\rho}$ on the stabilized

gradient, rather than by the (possibly unbounded) raw gradients. As a result, raw gradient explosions cannot induce a catastrophic single-step parameter jump. Moreover, on spike steps the update bound is further reduced by the factor $1/\kappa$, providing additional damping.

*Table 1.* **Performance of `GradientStabilizer` on FP4 and FP16 LLM pre-training.** Experiments are based on LLaMA-130M/350M. Validation perplexity is reported. The best results are in **bold**, and the second-best are underlined.

| Method | FP16 Training | | FP4 Training | |
| --- | --- | --- | --- | --- |
| | **130M** | **350M** | **130M** | **350M** |
| ADAM | 24.60 | 19.04 | 29.02 | 21.57 |
| + VALUE CLIP (Pascanu et al., 2013) | 24.48 | 19.01 | 28.95 | 20.48 |
| + NORM CLIP (Pascanu et al., 2013) | 24.17 | 18.84 | 28.31 | 22.29 |
| + AGC (Brock et al., 2021) | 24.32 | 18.76 | 28.52 | 19.24 |
| + ZCLIP (Kumar et al., 2025) | 24.16 | 18.62 | 28.34 | 19.23 |
| + GradientStabilizer | **23.32** | **17.83** | 26.82 | **18.89** |
| ADAMW | 24.31 | 19.21 | 28.72 | 21.35 |
| + VALUE CLIP (Pascanu et al., 2013) | 24.34 | 19.15 | 28.72 | 21.84 |
| + NORM CLIP (Pascanu et al., 2013) | 23.86 | 18.98 | 28.05 | 20.13 |
| + AGC (Brock et al., 2021) | 24.09 | 18.90 | 28.26 | 19.42 |
| + ZCLIP (Kumar et al., 2025) | 23.89 | 18.89 | 28.04 | 21.39 |
| + GradientStabilizer | **23.14** | **17.80** | 26.66 | **18.84** |
| **Training Tokens** | 2.2B | 6.6B | 2.2B | 6.6B |

# 4. Experiments

To demonstrate the effectiveness of the proposed method, we evaluate it on a diverse set of widely used tasks spanning LLM and quantization-aware pre-training, image classification, reinforcement learning, and time-series forecasting.

**Baselines.** We compare our method against several widely used gradient clipping approaches. **(1)** VALUE CLIP (Pascanu et al., 2013) clips each gradient element when its absolute value exceeds a predefined threshold. **(2)** NORM CLIP (Pascanu et al., 2013) rescales the entire gradient when its $\ell_2$ norm exceeds a threshold. **(3)** Adaptive Gradient Clipping (AGC) (Brock et al., 2021) clips gradients when the unit-wise ratio between the gradient norm and the parameter norm exceeds a threshold. **(4)** ZCLIP (Kumar et al., 2025) detects and clips abnormal gradients by computing $z$-score statistics of gradient norms.

**Models and Datasets.** For image classification, we evaluate on ImageNet-1K (Deng et al., 2009) using both transformer- and convolution-based architectures, including ViT-B (Dosovitskiy, 2020), ConvNeXt-T (Liu et al., 2022), and ResNet-50 (He et al., 2016). For language modeling, we train LLaMA-130M and LLaMA-350M (Touvron et al., 2023) on the C4 dataset. For reinforcement learning, we report results on the HalfCheetah-v4 environment. For time-series forecasting, we conduct experiments on the widely used Weather dataset using PatchTST (Nie, 2022).

**Hyperparameter Settings.** We use fixed hyperparameters across all experiments. For `GradientStabilizer`, we set $\gamma_1 = 0.6$ and $\gamma_2 = 0.999$. For VALUE CLIP and NORM

CLIP (Pascanu et al., 2013), we use thresholds 0.1 and 1.0, respectively. For AGC (Brock et al., 2021), we set the clipping factor to 0.01. For ZCLIP (Kumar et al., 2025), we follow the recommended defaults and set $\alpha = 0.97$ and the $z$-score threshold to 2.5. For a fair comparison, we *do not* tune hyperparameters for either the baselines or our method; the same settings are used throughout the paper.

## 4.1. Superior Performance of `GradientStabilizer`

**LLM Pre-training and Quantization-Aware Training.** We evaluate `GradientStabilizer` against VALUE CLIP, NORM CLIP, AGC, and ZCLIP in LLM pre-training under both FP16 and FP4 quantization-aware training. Experiments use LLaMA-130M and LLaMA-350M trained on C4 for 2.2B and 6.6B tokens. We consider ADAM and ADAMW as the base optimizers, and report validation perplexity (PPL) in Table 1. We observe that ❶ `GradientStabilizer` significantly improves final performance across model scales, for both Adam and AdamW base optimizers, under FP16 pre-training and FP4 quantization-aware training. For instance, on FP4-trained LLaMA-350M, `GradientStabilizer` reduces validation perplexity by approximately 2.5 PPL. ❷ With comparison to clipping baselines across settings, `GradientStabilizer` achieves the best performance among all the baselines. Among them, ZClip (Kumar et al., 2025), a recently proposed approach, typically yields the second-best results. We further observe larger gains under FP4 quantization-aware training than under FP16 training. A plausible explanation is that low-bit training is more prone to optimization instability due to quantization error (Panferov et al., 2025; Wortsman et al., 2023), in which case stabilizing the step magnitude provides greater benefit.

**ImageNet Classification.** To evaluate the effect of `GradientStabilizer` on vision task, we conduct experiments on ImageNet-1K using three standard architectures spanning transformers and CNNs: ViT-B, ConvNeXt-T, and ResNet-50. For ViT-B and ConvNeXt-T, we follow the official torchvision training recipes.[1][2] For ResNet-50, we use the same recipe as ConvNeXt-T. All methods are trained with the same setup for *120 epochs*. Top-1 accuracy (%) are reported in Table 2. We observe that `GradientStabilizer` consistently improves over the AdamW/Adam base optimizers across ViT-B, ConvNeXt-T, and ResNet-50. It obtains the best or second-best Top-1 accuracy in nearly all cases, yielding stable gains across diverse architectures. We also note that while ZCLIP is competitive on language models, it does not consistently out-

---

[1] https://github.com/pytorch/vision/tree/main/references/classification#vit_b_16
[2] https://github.com/pytorch/vision/tree/main/references/classification#convnext

*Table 2.* **Performance of GradientStabilizer on ImageNet-1K.** Top-1 accuracy (%) on ImageNet-1K using ViT-B, ConvNeXt-T, and ResNet-50 with AdamW/Adam baselines. The best results are in **bold**, and the second-best are underlined.

| Method | ViT-B | ConvNeXt-T | ResNet-50 |
|---|---|---|---|
| ADAMW | 79.3 | 79.6 | 77.3 |
| + VALUE CLIP (Pascanu et al., 2013) | 79.4 | 80.0 | 77.2 |
| + NORM CLIP (Pascanu et al., 2013) | 79.5 | 80.0 | 77.5 |
| + AGC (Brock et al., 2021) | 38.5 | 77.2 | **77.7** |
| + ZCLIP (Kumar et al., 2025) | 79.4 | **80.1** | 77.5 |
| + GradientStabilizer | **79.6** | 80.1 | 77.6 |
| ADAM | 77.1 | 77.6 | 75.5 |
| + VALUE CLIP (Pascanu et al., 2013) | 77.0 | 78.1 | 75.6 |
| + NORM CLIP (Pascanu et al., 2013) | 77.1 | 78.2 | 75.7 |
| + AGC (Brock et al., 2021) | 71.2 | 76.9 | **75.8** |
| + ZCLIP (Kumar et al., 2025) | 76.9 | 78.1 | 75.7 |
| + GradientStabilizer | **77.3** | 78.3 | 75.8 |
| Training Datasets | ImageNet-1K | ImageNet-1K | ImageNet-1K |

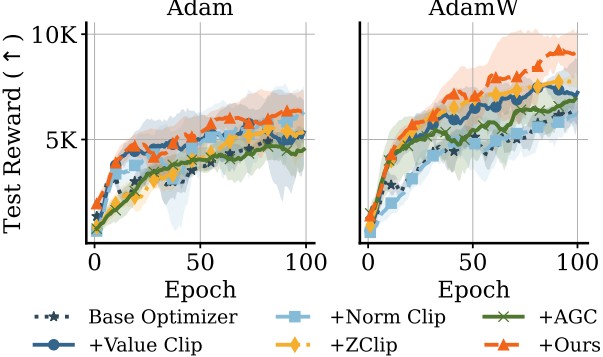

*Figure 3.* **Reinforcement learning on HalfCheetah-v4.** Mean episodic return ± standard deviation over $10\times$ evaluation rollouts, plotted against training epochs.

perform NORM CLIP on vision models, suggesting that its gains may be configuration-dependent.

**Reinforcement Learning.** We evaluate our proposed method on a continuous-control benchmark in MuJoCo. Specifically, we train the policy network using PPO (Schulman et al., 2017) on HALFCHEETAH-V4, following the standard training recipe from the Tianshou implementation.[3] Figure 3 reports the mean and standard deviation of the evaluation return computed over 10 rollouts. We observe that ❶ GradientStabilizer consistently attains the highest returns among all clipping baselines when combined with either Adam or AdamW, demonstrating robust gains across base optimizers; ❷ while standard clipping baselines generally improve over the unclipped optimizers, their relative ranking varies across settings, and no single clipping variant dominates consistently. The results for ANT-V4, HOPPER-V4 and WALKER-V4 are provided in Appendix C.4.

**Time Series Forecasting.** To evaluate the effectiveness of GradientStabilizer on time-series forecasting tasks, we conducted experiments on the Weather dataset by adopt-

---

[3] https://github.com/thu-ml/tianshou/tree/master/examples/mujoco

ing PatchTST (Nie, 2022), a widely used Transformer-based architecture, as the backbone model. We train PatchTST with Adam and AdamW, and compare against standard gradient-clipping baselines, following the official training recipes from the public PatchTST codebase.[4] Figure 5 reports the mean ± standard deviation over $10\times$ independent runs. We observe that ❶ GradientStabilizer yields substantial gains over the base optimizers and achieves the best performance among all clipping baselines; ❷ among clipping baselines, AGC is the strongest competitor, matching the leading performance, whereas VALUE CLIP provides no consistent improvement over the base optimizers.

**Summary**: ① *GradientStabilizer consistently delivers top-tier performance across diverse tasks and backbones, suggesting strong general applicability.* ② *In contrast, existing gradient clipping methods can be competitive in particular tasks or settings, but do not exhibit comparable consistency across diverse tasks.*

### 4.2. Stability Analysis

**Training stability.** We assess the effectiveness of GradientStabilizer by tracking the raw gradient norm (i.e., before applying GradientStabilizer) together with the training loss. Experiments are conducted for LLaMA-130M pre-training on C4 using ADAMW with learning rate $10^{-3}$. The results are shown in the two right subfigures of Figure 4. We observe that the base optimizer ADAMW suffer from a large number of gradient-norm spikes after several thousand steps where the norm exhibits frequent spikes occurring in bursts. These repeated spikes are accompanied by pronounced loss spikes, and the training dynamics eventually become unstable and diverge. In contrast, when combined with GradientStabilizer, the training loss curve remains well-behaved: the raw gradient norm does not exhibit the same spike bursts, and the loss continues to decrease smoothly without pronounced spike-induced instabilities. Refer to Appendix C.2 for comparisons with gradient clipping baselines.

**Learning rate stability.** We investigate the sensitivity of training stability to the learning rate when employing GradientStabilizer. Specifically, we perform a parameter sweep over learning rates in the interval $[10^{-4}, 3 \times 10^{-3}]$ and report the final validation loss. Experiments are conducted on LLaMA-130M pre-training using the $C4$ dataset, evaluating both Adam and AdamW optimizers with the FP4 training regime. The results, summarized in Figure 4 (right), demonstrate the stabilizing effect of

---

[4] https://github.com/yuqinie98/PatchTST/tree/main/PatchTST_supervised/scripts/PatchTST

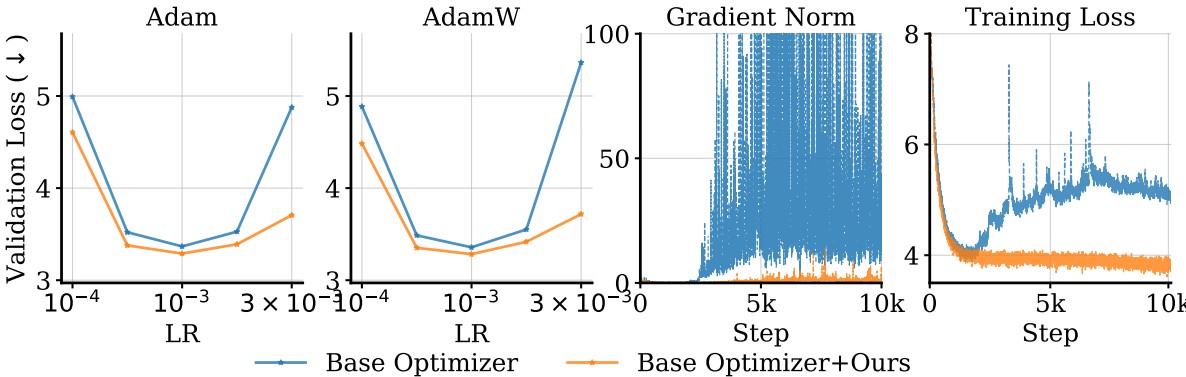

*Figure 4.* **Training and learning-rate stability.** *Left:* validation loss under different learning rates, illustrating learning-rate stability. *Right:* raw gradient norm (before `GradientStabilizer`) and training loss curves, illustrating training stability. All experiments are conducted on LLaMA-130M FP4 training on C4.

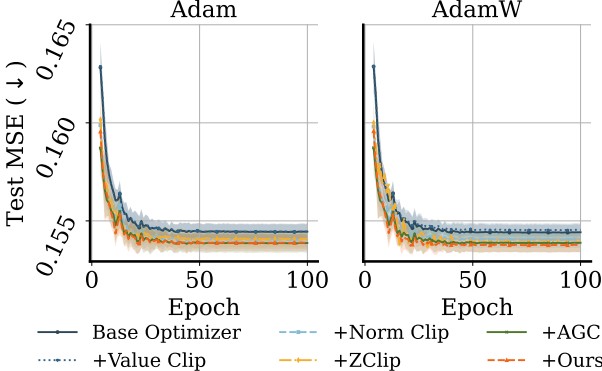

*Figure 5.* **Forecasting performance on Weather dataset.** Experiments are conducted on Weather using PatchTST with ADAM/ADAMW base optimizers. Results are reported as mean ± standard deviation over $10\times$ independent runs.

*Table 3.* **Weight-decay stability on ADAM.** Top-1 accuracy (%) of ViT-B under different weight-decay strengths using Adam on ImageNet-1K. The best results are in **bold**.

| Method | Weight Decay | | |
|---|---|---|---|
| | **0** | **1e-4** | **5e-4** |
| ADAM | 77.1 | 59.3 | 52.9 |
| + VALUE CLIP (Pascanu et al., 2013) | 77.0 | 60.2 | 29.2 |
| + NORM CLIP (Pascanu et al., 2013) | 77.1 | 53.4 | 20.6 |
| + AGC (Brock et al., 2021) | 71.2 | 0.1 | 0.1 |
| + ZCLIP (Kumar et al., 2025) | 76.9 | 59.9 | 30.5 |
| + `GradientStabilizer` | **77.3** | **78.7** | **72.4** |

our method. As the learning rate increases from $10^{-3}$ to $3 \times 10^{-3}$, the validation loss for GradientStabilizer combined with the base optimizer degrades more gracefully than that of the base optimizer alone, indicating superior robustness in high-learning-rate region. Conversely, in the low-learning-rate region, reducing the rate from $10^{-3}$ to $10^{-4}$ results in a more attenuated increase in loss when using GradientStabilizer. Overall, these findings suggest that GradientStabilizer effectively widens the effective operating range of learning rates. Refer to Appendix C.2 for comparisons with gradient clipping baselines.

**Weight Decay Stability on ADAM.** Prior work (Loshchilov & Hutter, 2017) has demonstrated that the Adam optimizer is highly sensitive to weight decay strength, as the effective decay is scaled by the second moment estimate and coupled with the learning rate. To investigate the impact of `GradientStabilizer` on this sensitivity, we conducted experiments on ImageNet-1K using the ViT-Base architecture. We swept the weight decay strength across a

range from 0 to $5 \times 10^{-4}$ and trained the models for 120 epochs. The results, summarized in Table 3, reveal that as weight decay strength increases, all baseline gradient clipping methods suffer substantial performance degradation compared to standard ADAM. This suggests that traditional gradient clipping exacerbates ADAM's sensitivity to weight-decay strength. In contrast, `GradientStabilizer` exhibits only minor performance drops and consistently outperforms ADAM, demonstrating that our method substantially mitigates this sensitivity. The results for ResNet-50 and ConvNeXt-T are provided in Appendix C.1.

### 4.3. Additional Analysis

**Performance under corrupted data.** Prior works (Shah et al., 2025; Liu & Ma, 2025; Talak et al., 2024) have established that corrupted inputs exacerbate training instability. To evaluate the efficacy of `GradientStabilizer` in mitigating these effects, we conduct experiments on a time series forecasting task using the Weather dataset. We employ the PatchTST architecture (Nie, 2022) with the ADAMW optimizer. To simulate data corruption, we inject Gaussian noise into the input $X$ with a probability of $p = 5\%$. The noise is sampled from $\mathcal{N}\left(0, \sigma^2\right)$, where the standard deviation is defined as $\sigma = \text{Noise\_Level} \cdot$

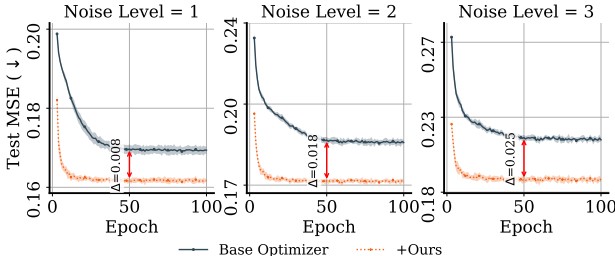

*Figure 6.* **Test MSE under input perturbations on Weather time-series data.** We corrupt $5\%$ of randomly selected input by adding zero-mean Gaussian noise. Specifically, the Gaussian noise is samples from $\mathcal{N}(0, \sigma^2)$, where the standard deviation is defined as $\sigma = \text{Noise\_Level} \cdot \max(X)$. Experiments are conducted with PatchTST using ADAMW optimizer and Test MSE is reported.

*Table 4.* **Combining GradientStabilizer with other optimizers.** Results are reported for Adam-Mini and Lion on LLaMA-130M/350M under FP4 training. The best results are in **bold**, and the second-best are underlined.

| Method | ADAM-MINI | | LION | |
|---|---|---|---|---|
| | **130M** | **350M** | **130M** | **350M** |
| Base Optimizer | 31.66 | 20.36 | 28.52 | 23.59 |
| + VALUE CLIP (Pascanu et al., 2013) | 31.52 | 20.99 | 28.32 | 23.20 |
| + NORM CLIP (Pascanu et al., 2013) | 30.54 | 21.03 | 28.23 | 22.98 |
| + AGC (Brock et al., 2021) | 30.41 | 21.84 | 28.33 | 23.20 |
| + ZCLIP (Kumar et al., 2025) | 30.50 | 21.14 | 28.16 | 23.13 |
| + GradientStabilizer | **27.75** | **18.64** | **26.72** | **22.75** |
| Training Tokens | 2.2B | 6.6B | 2.2B | 6.6B |

$\max(X)$ and Noise_Level control the corruption magnitude. Results in Figure 6 demonstrate that ADAM + GradientStabilizer significantly reduces the final Test MSE across all tested noise levels. Notably, the benefits of our method scale with the severity of the corruption; as the noise level increases from 1 to 3, the performance gain attributed to GradientStabilizer rises from approximately 0.008 to 0.025. This trend underscores the method's robustness in counteracting the adverse effects of input corruption. Comparisons with gradient clipping baselines are provided in Appendix C.3.

**Combined with other optimizers.** Our method is designed as an optimizer-agnostic gradient stabilizer. To further assess its generalizability, we evaluate it in conjunction with two recent optimizers: LION (Chen et al., 2023) and ADAM-MINI (Zhang et al., 2024). Experiments are conducted on LLaMA-130M and 350M models using FP4 training on the C4 dataset. The results in Table 4 demonstrate that applying GradientStabilizer to LION and ADAM-MINI yields significant improvements over the base optimizers. Furthermore, compared to standard gradient clipping baselines, our method achieves the largest and most consistent gains, underscoring its superior compatibility.

**Hyper-parameters Analysis.** GradientStabilizer introduces two hyperparameters, $\gamma_1$ and $\gamma_2$, which control the decay rates for the first and second moments of the

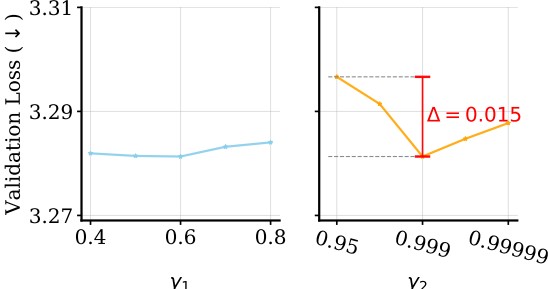

*Figure 7.* **Hyper-parameter analysis**. Experiments are conducted with FP4 training on LLaMA-130M and C4 with 2.2B tokens.

gradient norm. To investigate the sensitivity of the method to these parameters, we analyze the final validation loss while varying $\gamma_1 \in [0.5, 0.8]$ (fixing $\gamma_2 = 0.999$) and $\gamma_2 \in [0.95, 0.99999]$ (fixing $\gamma_1 = 0.6$). The results in Figure 7 indicate that while GradientStabilizer is relatively more sensitive to $\gamma_2$ than $\gamma_1$, it remains highly robust overall. Specifically, the maximum variation in validation loss across these broad ranges is less than 0.02, demonstrating that the method is stable across a wide configuration space.

## 5. Related Work

**Training Instability.** The training dynamics of deep neural networks are constrained by non-convex loss landscapes and pathological curvature, often leading to severe optimization difficulties (Martens et al., 2010). Early analyses by Bengio et al. (1994) and Glorot & Bengio (2010) identified the fundamental impediments of vanishing and exploding gradients, revealing that gradient norms can deviate exponentially with depth. In the context of Recurrent Neural Networks, Pascanu et al. (2013) demonstrated that error surfaces frequently exhibit steep cliffs, causing gradients to explode and weights to oscillate or diverge. Analogous instabilities persist in Reinforcement Learning (RL), where non-stationary data distributions and bootstrapping can induce value function degradations and even divergence (Sutton et al., 1998; Dasagi et al., 2019; Park et al., 2025). In the era of Large Language Models (LLMs), optimization instability manifests as sudden loss spikes that can reduce performance or even derail training entirely (Huang et al., 2025; Takase et al., 2023). Chowdhery et al. (2023) and Liu et al. (2025) report that scaling models beyond 60B parameters introduces critical gradient irregularities, often necessitating heuristic restart strategies or aggressive gradient clipping. This instability is exacerbated in low-bit training regimes, where the reduced dynamic range of quantized formats is ill-equipped to handle the heavy-tailed activation distributions inherent to large-scale models, leading to quantization-induced divergence (Wortsman et al., 2023; Panferov et al., 2025; Hao et al., 2025). To address these instabilities, researchers have developed various stabilization techniques. One prominent approach involves architectural and initial-

ization modifications, such as relocating LayerNorm (Xiong et al., 2020), inserting additional normalization layers after embeddings (Dettmers et al., 2021), and employing initialization schemes with reduced variance (Nguyen & Salazar, 2019). Other methods specifically target embedding dynamics by shrinking embedding gradients via reweighting (Ding et al., 2021; Zeng et al., 2022) or upscaling embeddings to stabilize LayerNorm gradients (Takase et al., 2023). A complementary line of work focuses on gradient constraints, utilizing clipping mechanisms that employ either soft scaling or hard truncation to suppress anomalous gradient magnitudes during backpropagation.

**Gradient Clipping Techniques.** To mitigate the risk of divergent optimization trajectories, clipping mechanisms constrain the magnitude of parameter updates. The prevailing paradigm is global gradient norm/value clip (Pascanu et al., 2013), which rescales the raw gradient vector whenever its $L_2$ norm/value exceeds a fixed hyperparameter. To address the sensitivity of fixed thresholds, adaptive variants have emerged: Adaptive Gradient Clipping (AGC) (Brock et al., 2021) and Clippy (Tang et al., 2023) dynamically modulate clipping thresholds relative to the norm of the model weights. More recently, Wang et al. (2025) and Kumar et al. (2025) proposed to identify and clip gradient anomalies by comparing the current gradient norm against a local moving average. In contrast to these anomaly-detection strategies, `GradientStabilizer` adaptively transforms the gradient norm without explicit outlier detection, enforcing intrinsic stability within the training dynamics.

## 6. Conclusion

In this paper, we introduced `GradientStabilizer`, an optimizer-agnostic gradient transformation that preserves update direction while regulating step magnitude via running norm statistics, offering a threshold-free alternative to heuristic clipping. Our theoretical analysis characterizes the variance-dampening properties of this approach in stationary regimes and establishes spike-invariant bounds, guaranteeing that transient gradient anomalies do not result in unbounded effective updates. Furthermore, we show that these bounds inherently regularize the moment states of Adam-style optimizers, ensuring well-defined dynamics per coordinate. Empirically, `GradientStabilizer` demonstrates superior stability across diverse tasks including low-precision LLM pre-training (FP16/FP4), ImageNet classification, RL, and time-series forecasting. By widening stable learning-rate regions and mitigating sensitivity to weight decay, our method offers a robust, drop-in solution for scaling deep learning optimization.

## Acknowledgements

The authors acknowledge the use of computing resources provided by the NVIDIA Academic Grant Program, the Isambard-AI National AI Research Resource (AIRR) and the Dutch national e-infrastructure, supported by the SURF Cooperative (Project EINF-17091).

## Impact Statement

The goal of this research is to advance the fundamental stability of neural network training. Optimization difficulties often act as a barrier to entry in deep learning, requiring significant computational budgets to tune hyperparameters and manage instabilities. By introducing a threshold-free mechanism that stabilizes training across diverse domains including LLMs, time-series forecasting, and RL, our work simplifies the training pipeline and broadens the stable operating region for optimizers. This improved reliability can help democratize access to large-scale model training for researchers with constrained computational resources. We do not foresee any direct negative societal consequences specific to this method, though we acknowledge the general dual-use nature of advancing deep learning performance.

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

# A. Proofs

## A.1. Proof of Lemma 3.2

*Proof.* Since $\mathbb{E}[R^2] = \text{Var}(R) + (\mathbb{E}[R])^2 = \sigma^2 + \mu^2$, we have

$$\rho_\star = \frac{\mu}{\sqrt{\mu^2 + \sigma^2}} = \frac{1}{\sqrt{1 + \sigma^2/\mu^2}} = \frac{1}{\sqrt{1 + c_v^2}} \leq 1.$$

$\square$

## A.2. Proof of Lemma 3.4

*Proof.* We analyze the behavior of the stabilized magnitude $\rho_t$ at a specific time step $t$ where a "gradient spike" occurs. Let the first and second moment estimators of the gradient norm $R_t$ follow the standard exponential moving average (EMA) updates with decay rates $\gamma_1$ and $\gamma_2$:

$$m_t^R = \gamma_1 m_{t-1}^R + (1 - \gamma_1)R_t, \tag{11}$$
$$v_t^R = \gamma_2 v_{t-1}^R + (1 - \gamma_2)R_t^2. \tag{12}$$

where $m_t^R >, v_t^R > 0$ since the gradient norm $R_t > 0$. Given the stabilized magnitude is defined by the ratio:

$$\rho_t = \frac{m_t^R}{\sqrt{v_t^R}} = \frac{\gamma_1 m_{t-1}^R + (1 - \gamma_1)R_t}{\sqrt{\gamma_2 v_{t-1}^R + (1 - \gamma_2)R_t^2}}. \tag{13}$$

Since $v_{t-1}^R \geq 0$ and $\gamma_2 \geq 0$, therefore, we can lower bound the term inside the square root by ignoring the historical second moment:

$$\sqrt{\gamma_2 v_{t-1}^R + (1 - \gamma_2)R_t^2} \geq \sqrt{(1 - \gamma_2)R_t^2} = R_t\sqrt{1 - \gamma_2}. \tag{14}$$

Applying this lower bound to the denominator yields an upper bound for $\rho_t$:

$$\rho_t \leq \frac{\gamma_1 m_{t-1}^R + (1 - \gamma_1)R_t}{R_t\sqrt{1 - \gamma_2}} = \frac{\gamma_1}{\sqrt{1 - \gamma_2}}\left(\frac{m_{t-1}^R}{R_t}\right) + \frac{1 - \gamma_1}{\sqrt{1 - \gamma_2}}.. \tag{15}$$

The lemma assumes a spike condition $R_t \geq \kappa m_{t-1}^R$ for some threshold $\kappa \gg 1$. We can rearrange this inequality to bound the ratio of the historical moment to the current norm:

$$\frac{m_{t-1}^R}{R_t} \leq \frac{1}{\kappa}. \tag{16}$$

Substituting this inequality into the expression for $\rho_t$, we have:

$$\rho_t \leq \frac{1 - \gamma_1}{\sqrt{1 - \gamma_2}} + \frac{\gamma_1}{\kappa\sqrt{1 - \gamma_2}}. \tag{17}$$

$\square$

## A.3. Proof of Lemma 3.6

*Proof.* With $m_0^{(R)} = v_0^{(R)} = 0$, unrolling the EMA recursions yields

$$m_t^{(R)} = (1 - \gamma_1)\sum_{k=1}^t \gamma_1^{t-k} R_k, \tag{18}$$

$$v_t^{(R)} = (1 - \gamma_2)\sum_{k=1}^t \gamma_2^{t-k} R_k^2. \tag{19}$$

Let $w_k = \gamma_2^{t-k} > 0$. By Cauchy–Schwarz inequality,

$$
\begin{aligned}
\sum_{k=1}^{t} \gamma_1^{t-k} R_k = \sum_{k=1}^{t} \Big(\frac{\gamma_1^{t-k}}{\sqrt{w_k}}\Big)\Big(\sqrt{w_k} R_k\Big) &\leq \Big(\sum_{k=1}^{t} \frac{\gamma_1^{2(t-k)}}{w_k}\Big)^{\frac{1}{2}} \Big(\sum_{k=1}^{t} w_k R_k^2\Big)^{\frac{1}{2}} \\
&= \Big(\sum_{j=0}^{t-1} \Big(\frac{\gamma_1^2}{\gamma_2}\Big)^{j}\Big)^{\frac{1}{2}} \Big(\sum_{k=1}^{t} \gamma_2^{t-k} R_k^2\Big)^{\frac{1}{2}} \\
&= \frac{\sqrt{1 - (\gamma_1^2/\gamma_2)^t}}{\sqrt{1 - \gamma_1^2/\gamma_2}} \Big(\sum_{k=1}^{t} \gamma_2^{t-k} R_k^2\Big)^{\frac{1}{2}},
\end{aligned}
\tag{20}
$$

since $\gamma_1^2 < \gamma_2$ so $\sqrt{1 - (\gamma_1^2/\gamma_2)^t} \leq 1$, then we have:

$$
\sum_{k=1}^{t} \gamma_1^{t-k} R_k \leq \frac{\sqrt{1 - (\gamma_1^2/\gamma_2)^t}}{\sqrt{1 - \gamma_1^2/\gamma_2}} \Big(\sum_{k=1}^{t} \gamma_2^{t-k} R_k^2\Big)^{\frac{1}{2}}
$$

Combining (18)–(20) and using (19), we obtain

$$
m_t^{(R)} \leq (1 - \gamma_1) \cdot \frac{1}{\sqrt{1 - \gamma_1^2/\gamma_2}} \Big(\sum_{k=1}^{t} \gamma_2^{t-k} R_k^2\Big)^{\frac{1}{2}} = \frac{1 - \gamma_1}{\sqrt{1 - \gamma_2}} \cdot \frac{1}{\sqrt{1 - \gamma_1^2/\gamma_2}} \sqrt{v_t^{(R)}}.
$$

Dividing both sides by $\sqrt{v_t^{(R)}}$ yields

$$
\rho_t = \frac{m_t^{(R)}}{\sqrt{v_t^{(R)}}} \leq \frac{1 - \gamma_1}{\sqrt{1 - \gamma_2}} \cdot \frac{1}{\sqrt{1 - \gamma_1^2/\gamma_2}} = \bar{\rho}.
$$

The proof is complete. ☐

### A.4. Proof of Proposition 3.8

*Proof.* Fix any $t \geq 1$. By the conclusion of Lemma 3.6, we have $\|\tilde{g}_t\|_2 \leq \bar{\rho}$. Moreover, for any vector $x \in \mathbb{R}^d$ it holds that $\|x\|_\infty \leq \|x\|_2$ since for each coordinate $i$, $|x_i| \leq \sqrt{\sum_{j=1}^{d} x_j^2} = \|x\|_2$. Applying this inequality to $x = \tilde{g}_t$ yields

$$
\|\tilde{g}_t\|_\infty \leq \|\tilde{g}_t\|_2 \leq \bar{\rho}.
$$

Finally, $\|\tilde{g}_t\|_\infty = \max_i |\tilde{g}_{t,i}|$ implies $|\tilde{g}_{t,i}| \leq \bar{\rho}$ for all $i$, completing the proof. ☐

### A.5. Proof of Theorem 3.9

*Proof.* Fix any coordinate $i \in [d]$. By Proposition 3.8, we have $|\tilde{g}_{t,i}| \leq \bar{\rho}$ for all $t \geq 1$.

**Bound on $m_t$.** The base case holds since $m_{0,i} = 0 \leq \bar{\rho}(1 - \beta_1^0)$. For $t \geq 1$, using (5) and the triangle inequality,

$$
|m_{t,i}| = |\beta_1 m_{t-1,i} + (1 - \beta_1)\tilde{g}_{t,i}| \leq \beta_1 |m_{t-1,i}| + (1 - \beta_1)|\tilde{g}_{t,i}| \leq \beta_1 |m_{t-1,i}| + (1 - \beta_1)\bar{\rho}.
$$

Unrolling the inequality from $m_{0,i} = 0$ yields

$$
|m_{t,i}| \leq (1 - \beta_1)\bar{\rho} \sum_{k=0}^{t-1} \beta_1^k = \bar{\rho}(1 - \beta_1^t).
$$

Taking the maximum over $i$ gives $\|m_t\|_\infty \leq \bar{\rho}(1 - \beta_1^t) \leq \bar{\rho}$.

**Bound on $v_t$.** Since $v_{0,i} = 0$ and $\tilde{g}_{t,i}^2 \geq 0$, the recursion (6) implies $v_{t,i} \geq 0$ for all $t$. Moreover, by (6) and $|\tilde{g}_{t,i}| \leq \bar{\rho}$,

$$v_{t,i} = \beta_2 v_{t-1,i} + (1 - \beta_2)\tilde{g}_{t,i}^2 \leq \beta_2 v_{t-1,i} + (1 - \beta_2)\bar{\rho}^2.$$

Unrolling from $v_{0,i} = 0$ yields

$$v_{t,i} \leq (1 - \beta_2)\bar{\rho}^2 \sum_{k=0}^{t-1} \beta_2^k = \bar{\rho}^2(1 - \beta_2^t).$$

Taking the maximum over $i$ gives $\|v_t\|_\infty \leq \bar{\rho}^2(1 - \beta_2^t) \leq \bar{\rho}^2$.

**Bound on $\hat{v}_t$ (AMSGrad).** For AMSGrad, $\hat{v}_t$ is defined elementwise by (7), with $\hat{v}_{0,i} = 0$. Thus $\hat{v}_{t,i} = \max\{\hat{v}_{t-1,i}, v_{t,i}\} = \max_{1 \leq s \leq t} v_{s,i}$. Using the bound on $v_{s,i}$ above and the fact that $1 - \beta_2^s$ is nondecreasing in $s$,

$$\hat{v}_{t,i} \leq \max_{1 \leq s \leq t} \bar{\rho}^2(1 - \beta_2^s) = \bar{\rho}^2(1 - \beta_2^t).$$

Taking the maximum over $i$ yields $\|\hat{v}_t\|_\infty \leq \bar{\rho}^2(1 - \beta_2^t) \leq \bar{\rho}^2$. Combining the three bounds completes the proof. $\square$

### A.6. Bounded Per-step Parameter Change for Adam/AMSGrad

**Corollary A.1** (Bounded Per-step Parameter Change under `GradientStabilizer` for Adam/AMSGrad ). *Consider Adam/AMSGrad updates with an explicit $\epsilon > 0$ in the denominator:*

$$\theta_{t+1} = \theta_t - \alpha_t \frac{m_t}{\sqrt{\hat{v}_t} + \epsilon}, \tag{21}$$

*where division and square-root are elementwise and $\hat{v}_t = v_t$ for Adam. Under the conditions of Theorem 3.9, with $\beta_1, \beta_2 \in [0, 1)$, for all $t \geq 1$ and all coordinates $i \in [d]$,*

$$|\theta_{t+1,i} - \theta_{t,i}| \leq \alpha_t \frac{\bar{\rho}(1 - \beta_1^t)}{\sqrt{\hat{v}_{t,i}} + \epsilon}. \tag{22}$$

*Proof.* From the update rule (21) in main text, all operations are coordinatewise, so for any $i \in [d]$,

$$\theta_{t+1,i} - \theta_{t,i} = -\alpha_t \frac{m_{t,i}}{\sqrt{\hat{v}_{t,i}} + \epsilon}.$$

Taking absolute values gives

$$|\theta_{t+1,i} - \theta_{t,i}| = \alpha_t \frac{|m_{t,i}|}{\sqrt{\hat{v}_{t,i}} + \epsilon}.$$

By Theorem 3.9, we have the tight first-moment bound $\|m_t\|_\infty \leq \bar{\rho}(1 - \beta_1^t)$, and hence $|m_{t,i}| \leq \bar{\rho}(1 - \beta_1^t)$ for every coordinate $i$. Substituting this inequality into the display above yields

$$|\theta_{t+1,i} - \theta_{t,i}| \leq \alpha_t \frac{\bar{\rho}(1 - \beta_1^t)}{\sqrt{\hat{v}_{t,i}} + \epsilon},$$

which proves the claim. $\square$

*Remark* A.2. Corollary A.1 shows that, under `GradientStabilizer`, each coordinate update is controlled by the *intrinsic* bound $\bar{\rho}$ on the stabilized gradient magnitude, rather than by the (possibly unbounded) raw gradients. As a result, raw gradient explosions cannot induce a catastrophic single-step parameter jump, ensuring training stability even in the presence of arbitrarily large gradient spikes.

## A.7. Proof of Corollary 3.11

*Proof.* Recall the update rule from Corollary 3.11, we have

$$\theta_{t+1} - \theta_t = -\eta_t \tilde{g}_t,$$

Under Proposition 3.8 and thus

$$\|\theta_{t+1} - \theta_t\|_\infty = \eta_t \|\tilde{g}_t\|_\infty \leq \eta_t \|\tilde{g}_t\|_2,$$

Under the conditions of Lemma 3.6, $\|\tilde{g}_t\|_2 \leq \bar{\rho}$ for all $t \geq 1$, which yields

$$\|\theta_{t+1} - \theta_t\|_\infty \leq \eta_t \bar{\rho}.$$

Moreover, on spike steps satisfying $R_t \geq \kappa\, m_{t-1}^{(R)}$, Lemma 3.4 gives

$$\|\tilde{g}_t\|_2 \leq \frac{1 - \gamma_1}{\sqrt{1 - \gamma_2}} + \frac{\gamma_1}{\kappa \sqrt{1 - \gamma_2}},$$

and therefore implies

$$\|\theta_{t+1} - \theta_t\|_\infty \leq \eta_t \left( \frac{1 - \gamma_1}{\sqrt{1 - \gamma_2}} + \frac{\gamma_1}{\kappa \sqrt{1 - \gamma_2}} \right).$$

$\square$

# B. Experimental Details for Reproducibility

## B.1. Codes for gradient clipping baselines

We adopt the official code for the gradient clipping baselines.

- AGC:https://github.com/huggingface/pytorch-image-models/blob/main/timm/utils/agc.py

- ZCLIP:https://github.com/bluorion-com/ZClip

- NORM CLIP:https://github.com/pytorch/pytorch/blob/v2.10.0/torch/nn/utils/clip_grad.py#L185

- VALUE CLIP: https://github.com/pytorch/pytorch/blob/v2.10.0/torch/nn/utils/clip_grad.py#L257

Table 5 summarizes the hyper-parameter settings for our method and all baselines; these values are fixed across all experiments in the main paper.

*Table 5.* Hyper-parameter settings for GradientStabilizer and clipping baselines (fixed across all experiments).

| Method | Hyper-parameters |
|---|---|
| GradientStabilizer | $\gamma_1 = 0.6$, $\gamma_2 = 0.999$ |
| VALUE CLIP (Pascanu et al., 2013) | threshold $= 0.1$ |
| NORM CLIP (Pascanu et al., 2013) | threshold $= 1.0$ |
| AGC (Brock et al., 2021) | clipping factor $= 0.01$ |
| ZCLIP (Kumar et al., 2025) | $\alpha = 0.97$, $z$-score threshold $= 2.5$ |

*Table 6.* Training hyper-parameters for LLaMA models under Adam and AdamW.

| Model | Optimizer | LR | Batch-size | Total batch-size | Warmup | Steps |
|-------|-----------|-----|-----------|------------------|--------|-------|
| LLaMA-130M | Adam | 1e-3 | 128 | 512 | 2000 | 20000 |
| | AdamW | 1e-3 | 128 | 512 | 2000 | 20000 |
| LLaMA-350M | Adam | 1e-3 | 128 | 512 | 2000 | 60000 |
| | AdamW | 1e-3 | 128 | 512 | 2000 | 60000 |

*Weight decay:* Adam uses 0; AdamW uses 0.01.

### B.2. Training configurations for Experiments in main content

**LLM-Pretraining and Quantization-Aware Trainig:** We implement our training framework based on the GaLore codebase `https://github.com/jiaweizzhao/GaLore`. For Quantization-Aware Training (QAT), we employ FP4 bit (E1M2 format: 1-bit exponent, 2-bit mantissa). Specifically, we quantize all weights and activations to 4-bit floating point (FP4) precision. Table 6 shows the details of hyper-parameter settings for the training.

**ImageNet-1K Classification.** We train ConvNeXt-T, ResNet-50, ViT-B using the default training scripts provided by torchvision but with 120 epochs only due to the limiation of our computing resources.

- ConvNeXt-T:`https://github.com/pytorch/vision/tree/main/references/classification#convnext`.

- ViT-B:`https://github.com/pytorch/vision/tree/main/references/classification#vit_b_16`

- ResNet-50: we use the same training script as ConvNeXt-T. `https://github.com/pytorch/vision/tree/main/references/classification#convnext`.

**Reinforcement Learning.** We are training the policy network for the MuJoCo environment using the training scripts provided by the Tianshou framework `https://github.com/thu-ml/tianshou/tree/master/examples/mujoco`. Table 7 shows the hyper-parameter settings for reinforcement learning experiments.

*Table 7.* Optimizer hyper-parameters for MuJoCo policy training (Tianshou).

| Optimizer | Learning rate | Weight decay |
|-----------|---------------|--------------|
| Adam | $5 \times 10^{-4}$ | 0 |
| AdamW | $5 \times 10^{-4}$ | $1 \times 10^{-2}$ |

**Time Series Forescasting.** We trains PatchTST based on their official public code and scripts `https://github.com/yuqinie98/PatchTST/blob/main/PatchTST_supervised/scripts/PatchTST/weather.sh`. Table 8 shows the hyper-parameter settings for the TSF task.

*Table 8.* Optimizer hyper-parameters for Time Series Forescasting task.

| Optimizer | Learning rate | Weight decay |
|-----------|---------------|--------------|
| Adam | $1 \times 10^{-4}$ | 0 |
| AdamW | $1 \times 10^{-4}$ | $1 \times 10^{-2}$ |

## C. Additional Experiments

### C.1. Weight Decay Stability on ADAM for ResNet-50 and ConvNeXt-T.

The results in Table 9 demonstrate that as weight decay strength varies from 0 to $10^{-4}$, ADAM and all gradient clipping baselines experience a substantial drop in performance. In contrast, `GradientStabilizer` not only avoids this degradation but yields improved performance, highlighting its effectiveness in mitigating ADAM's sensitivity to weight decay.

*Table 9.* **Weight-decay stability on Adam.** Top-1 accuracy (%) of ConvNeXt-T and ResNet-50 under different weight-decay strengths using Adam on ImageNet-1K. The best results are in **bold**.

| Method | ConvNeXt-T | | ResNet-50 | |
|---|---|---|---|---|
| | **WD=0** | **WD=1e-4** | **WD=0** | **WD=1e-4** |
| Base Optimizer | 77.6 | 35.2 | 75.5 | 69.8 |
| + Value Clip (Pascanu et al., 2013) | 78.1 | 15.7 | 75.6 | 69.8 |
| + Norm Clip (Pascanu et al., 2013) | 78.2 | 27.2 | 75.7 | 64.5 |
| + AGC (Brock et al., 2021) | 76.9 | 2.6 | 75.8 | 64.2 |
| + Zclip (Kumar et al., 2025) | 78.1 | 26.4 | 75.8 | 64.5 |
| + GradientStabilizer | **78.3** | **78.8** | **75.8** | **76.6** |
| Training Dataset | ImageNet-1K | ImageNet-1K | ImageNet-1K | ImageNet-1K |

### C.2. Comparison with Gradient Clipping baselines on Learning rate and Training Stability.

**Learning rate stability.** We compare learning rate stability against gradient clipping baselines in Figure 8 (Left). The results demonstrate that GradientStabilizer exhibits superior stability compared to all baselines across both low and high learning rate regimes. Additionally, we observe that ZCLIP and NORM CLIP achieve comparable stability in high learning rate regions.

**Training stability.** Figure 8 (Right) compares the training stability of our method against gradient clipping baselines. The results demonstrate that GradientStabilizer maintains lower gradient norms and yields smoother training loss curves than the baselines, highlighting its effectiveness in stabilizing training dynamics.

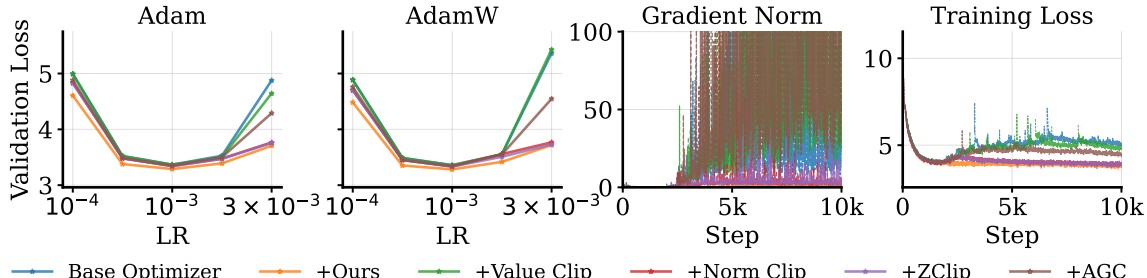

*Figure 8.* **Training and learning-rate stability.** *Left:* validation loss under different learning rates, illustrating learning-rate stability. *Right:* raw gradient norm (before GradientStabilizer) and training loss curves, illustrating training stability. All experiments are conducted on LLaMA-130M FP4 training on C4.

### C.3. Comparison with Gradient Clipping baselines on Corrupted Data.

Figure 9 compares our method against gradient clipping baselines on corrupted data, using the same experimental setup as Figure 6. GradientStabilizer consistently outperforms or matches baselines across all noise levels. Notably, AGC and ZCLIP also demonstrate competitive performance.

### C.4. Experiments on RL Tasks with Additional Environments

We further evaluate our method on the ANT-V4, HOPPER-V4, and WALKER2D-V4 environments using the AdamW optimizer. The results presented in Figure 10 demonstrate that GradientStabilizer consistently achieves the highest rewards among all gradient clipping baselines across these tasks. Notably, while we observe that standard gradient clipping baselines suffer from performance degradation compared to the base optimizer on WALKER2D-V4, GradientStabilizer effectively maintains performance parity with the unclipped baseline.

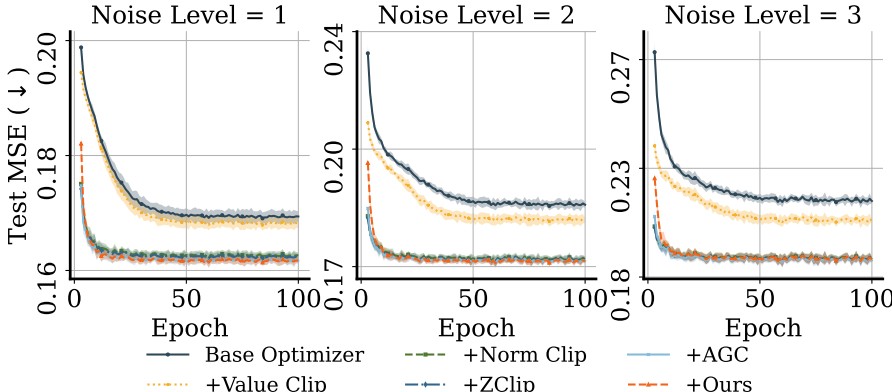

*Figure 9.* **Test MSE under input perturbations on Weather time-series data.** We corrupt $5\%$ of randomly selected input by adding zero-mean Gaussian noise. Specifically, the Gaussian noise is samples from $\mathcal{N}\left(0, \sigma^2\right)$, where the standard deviation is defined as $\sigma = \text{Noise\_Level} \cdot \max(X)$. Experiments are conducted with PatchTST using AdamW optimizer and Test MSE is reported.

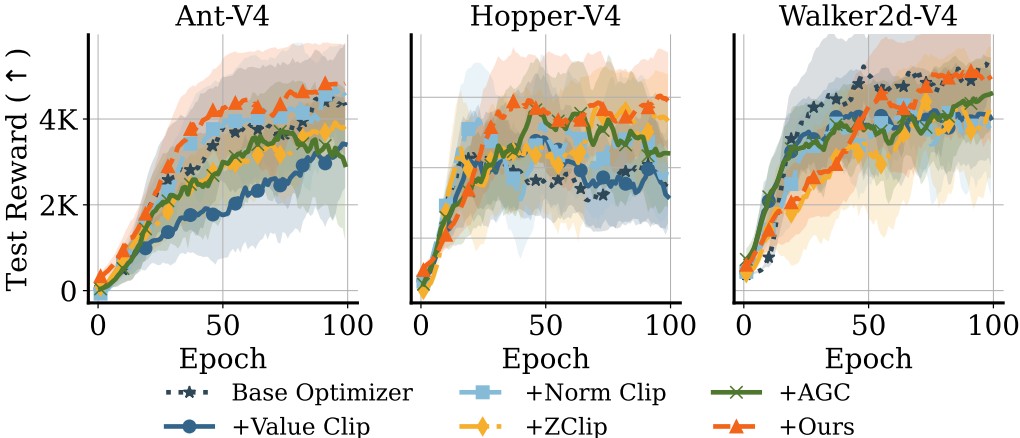

*Figure 10.* **Reinforcement learning on Ant-V4, Hopper-V4 and Walker-V4.** Mean episodic return $\pm$ standard deviation over $10\times$ evaluation rollouts, plotted against training epochs.

