# OpenReview forum: "GradientStabilizer: Fix the Norm, Not the Gradient"
_ICML.cc/2026/Conference — ICML 2026 regular_

### Official Review · Reviewer_1jTr · 2026-02-27

**Soundness:** 3
**Presentation:** 3
**Significance:** 3
**Originality:** 3
**Overall Recommendation:** 4
**Confidence:** 4

**Summary:**

This paper proposes *GradientStabilizer*, a simple gradient transform meant to make training robust to rare gradient-norm spikes. The method keeps the instantaneous gradient direction but replaces its magnitude with a stabilized estimate computed from EMA statistics of the gradient norm and its square, effectively using a mean-to-RMS style ratio to set the update size before passing the transformed gradient to a base optimizer (e.g., Adam/AdamW). The paper provides (i) a theoretical analysis showing that the resulting effective update magnitude is bounded on spike steps (independent of the raw spike size), and (ii) experiments across low-precision LLM pre-training (FP16/FP4), ImageNet classification, MuJoCo RL, and time-series forecasting.

An important concept considered by the article is the explicit decoupling of gradient *direction* and *step magnitude* as a stability mechanism under heavy-tailed / spiky norm behavior. An important domain examined by the manuscript is low-precision LLM pre-training, where the paper reports noticeable perplexity improvements and clearer stability gains than in some of the other benchmarks.

**Compliance With Llm Reviewing Policy:**

Affirmed.

**Final Justification:**

Thanks for the detailed rebuttal. This significantly improves the paper.

The additional comparisons against tuned NormClip and ZClip are helpful and strengthen the empirical case, especially given the sensitivity of clipping to threshold choice. The new mechanism-isolation ablations are also valuable and address one of my main concerns: they make it clearer that the specific combination used in GS is doing more than generic normalization or simple magnitude smoothing. I also appreciate the added diagnostics in RL, where the spike suppression and performance gains are quite compelling.

That said, some aspects remain less clear. The comparison to strong baselines is still somewhat limited in scope (e.g., not all domains are covered, and AGC is not included), and the robustness story outside LLMs remains uneven. In particular, the ImageNet results still do not fully support the spike-driven motivation, since catastrophic spikes do not seem to occur there. I also think some of the diagnostic evidence could be more complete (e.g., across seeds or with more detailed trajectories), although I understand this is hard to fully address within the rebuttal.

Overall, the rebuttal resolves a substantial part of my concerns and improves my confidence in the method, even if some questions would require more extensive revisions. I will increase my score to 4 (borderline accept).

**Key Questions For Authors:**

1. **How does GradientStabilizer compare to *tuned* norm clipping (and tuned AGC/ZClip) on each domain?** If you add a small tuning budget for clipping thresholds, do the gaps close, or do you still see consistent stability/performance benefits?
2. **Can you provide domain-level robustness diagnostics beyond LLMs?** For ImageNet and RL, please include at least (a) gradient/update norm trajectories (or distributions), (b) divergence or collapse rates across seeds, and (c) how often spikes occur and how strongly the method rescales them.
3. **Can you add mechanism-isolation baselines?** In particular, compare to (i) pure normalized-gradient updates (ρt ≡ 1), and (ii) a one-moment magnitude rule (ρt based only on an EMA of ∥g∥). This would clarify what part of the recipe is doing the work.
4. **What is the typical range of ρt during training in your main settings (median/percentiles), and how does that compare to the effective scaling induced by clipping?** A simple plot could make the “this is not just clipping in disguise” argument much stronger.

**Limitations:**

Yes

**Strengths And Weaknesses:**

### Soundness

**Strengths**
- The method is clean and well-specified: keep gradient direction, replace magnitude with a stabilized estimate from EMA statistics of the gradient norm and its square, then feed the transformed gradient into a standard optimizer. It’s easy to reason about and easy to implement.
- The theoretical framing matches the motivation: the transform bounds the effective update on spike steps and helps argue why optimizer states (e.g., Adam moments) are less likely to get corrupted by rare large-norm gradients.
- The paper includes robustness-style analyses (LR / weight decay sweeps and a hyperparameter sensitivity study), which helps with credibility beyond a single tuned setup.

**Weaknesses**
- The main algorithmic claim implicitly leans toward “you can replace gradient clipping”, but the evidence against *strong* clipping baselines is not decisive. Avoiding threshold tuning is a nice usability point, yet it also weakens the “strictly better than clipping” story unless tuned clipping (or adaptive clipping with a reasonable tuning budget) is explicitly compared across domains.
- Mechanism isolation is incomplete: it’s hard to tell how much of the gain comes from the specific mean-to-RMS style magnitude rule versus a more generic “normalize direction + controlled step size” effect. A couple of ablations/baselines would make the causal story much stronger.

---

### Presentation

**Strengths**
- The paper is generally easy to follow: motivation, method, and how it plugs into existing optimizers are clearly described.
- The narrative around spikes and stability is coherent, and the additional robustness plots (LR region, weight decay) make the presentation feel more practically grounded than papers that only show a single final number.

**Weaknesses**
- For the vision and RL sections, the presentation under-serves the paper’s own motivation. If the story is about spike-driven instability, showing mostly final test accuracy/return (with results that are often very close) is not very persuasive. The paper would read stronger if it consistently reported stability diagnostics (gradient/update norm traces, divergence rate across seeds, spike frequency) outside the LLM setting too.

---

### Significance

**Strengths**
- If the method reliably reduces divergence and widens the stable training region without threshold tuning, that is immediately useful. Clipping is everywhere and threshold selection is a recurring pain point.
- The low-precision LLM experiments are the most compelling part of the paper: that’s a setting where stability problems are real and expensive, so improvements there matter.

**Weaknesses**
- In several non-LLM benchmarks the headline metrics are nearly tied. Without stronger robustness evidence, it’s hard to justify switching from the “known default” (clipping) purely on practical grounds.
- The paper doesn’t yet make the case that the benefits generalize as *robustness improvements* across domains rather than being primarily an LLM/low-precision story.

---

### Originality

**Strengths**
- The idea is simple but genuinely nice: normalize away the norm, then re-inject a statistically stabilized magnitude using two norm moments in an Adam-like ratio. It’s a clever, practical twist on controlling step magnitudes without a hard clipping threshold.
- The “fix the norm, not the gradient” framing is a helpful perspective that could influence how people think about spike handling.

**Weaknesses**
- Conceptually adjacent to existing families of methods (clipping, normalized updates, adaptive scaling), so the novelty is more in the specific construction and framing than in introducing an entirely new optimization paradigm. That’s fine, but it raises the bar on experiments: the paper needs clearer evidence that this particular construction beats strong versions of the nearby baselines.

---

> ### Author Rebuttal · Authors · 2026-03-31
>
> Note: **GS denotes GradientStabilizer in this rebuttal**.
>
> **Q1: How does GS compare to tuned norm clipping... ?**
>
> **A1:** We conducted new experiments with individually tuned thresholds for NormClip and ZClip across two tasks. GS consistently outperforms both tuned baselines.
>
> - Moreover, the clipping baselines are highly sensitive to threshold choice (e.g., NormClip on FP16: 23.86–25.20), while GS is substantially less sensitive to hyperparameter selection (Figure 6 in submission).
>
> **FP16 Training**
>
> | Method |Threshold|Result|
> |---|---|---|
> | **GS** |—|**23.32**|
> | NormClip |0.5|24.23|
> | NormClip |1.0| 23.86|
> | NormClip |1.5|24.01|
> | NormClip |2.0|25.20|
> | ZClip |1.5|24.62|
> | ZClip |2.5|23.89|
> | ZClip |3.0|23.82|
> | ZClip |4.0|24.63|
>
> **ViT-B on ImageNet**
> | Method | Threshold | Result |
> |---|---|---|
> | **GS** | — | **79.60** |
> | NormClip | 0.5 | 79.41 |
> | NormClip | 1.0 | 79.50 |
> | NormClip | 1.5 | 79.40 |
> | NormClip | 2.0 | 79.35 |
> | ZClip | 1.0 | 79.48 |
> | ZClip | 1.5 | 79.54 |
> | ZClip | 2.5 | 79.40 |
> | ZClip | 3.0 | 79.38 |
>
>
> **Q2: Can you provide domain-level robustness diagnostics beyond LLMs?...**
>
> **A2:** We provide domain-level robustness diagnostics for both RL and ImageNet beyond final performance here.
>
> **(a) Gradient-norm distributions.**
> We report statistics of the global gradient norm in the table below.
>
> - **RL (HalfCheetah):** AdamW suffers from highly out-of-distribution gradient norms (mean 126.7, std 106.84, max 7567.21), whereas AdamW + GS is dramatically more concentrated (3.04 / 1.19 / 33.5), strongly suppressing rare but catastrophic spikes.
> - **ImageNet (ConvNeXt-T):** GS does *not* simply shrink all gradients. The average norm is actually larger (13.7 vs. 3.28), while variability and extreme values are reduced (std 2.51→1.39, max 80.07→44.9). This is consistent with *stabilization* mechanism: preserving healthy update magnitudes while reducing tail instability.
>
> | |Mean|Std|Max|
> |---|---|---|---|
> | **RL-HalfCheetah**| | | |
> | AdamW |126.7|106.84|7567.2|
> | AdamW+GS | 3.04| 1.2| 33.5|
> | **ConvNeXt-T-ImageNet** | | | |
> | AdamW |3.28|2.51|80.07|
> | AdamW+GS |13.7|1.39|44.9|
>
> **(b) Divergence or collapse across seeds.**
> Under standard training recipes, we did not observe complete divergence or collapse in either domain. However, in RL, AdamW frequently exhibits sharp gradient norm spikes, whereas these gradient spikes can be significantly reduced by applying GS.
>
> **(c) Spike frequency and rescaling strength.**
> Following the spike definition in our paper, we count a gradient spike whenever $R_t \geq \kappa m^{R}_{t-1}$ with $\kappa=10$. Under this criterion, AdamW exhibits 36 spike events in RL, whereas AdamW+GS exhibits 0; correspondingly, the mean test reward improves from 6601 to 9537. In ImageNet, no catastrophic spikes are observed for either method.
>
> | | # Spikes |Performance|
> |---|---|---|
> | **RL-HalfCheetah** | |*Test Reward* |
> | AdamW | 36 | 6601|
> | AdamW+GS | 0 |9537|
> | **ConvNeXt-T** | |*Acc*|
> | AdamW |0|79.6|
> | AdamW+GS |0|80.1|
>
> **Q3:Can you add mechanism-isolation baselines?**
>
> **A3:** We have added both mechanism-isolation baselines: (i)  $\rho_t$ ≡ 1, and (ii)  $ρ_t$ is given only by an EMA of $‖g_t‖$.
>
> Both ablations slightly improve over the base optimizer, indicating that direction-preserving normalization and magnitude smoothing are each useful. However, neither matches our GS, indicating GS yields a more reliable magnitude estimate and better stabilizes training.
>
> **Table: Ablation of GS Components**
>
> | Method | FP16 (↓) | FP4 (↓) |
> |---|---|---|
> |(i)| 24.13 | 28.01 |
> |(ii), γ₁=0.6 | 24.23 | 28.66 |
> |(ii), γ₁=0.8 | 24.15 | 28.42 |
> |(ii), γ₁=0.9 | 24.10 | 28.46 |
> | **GS** | **23.14** | **26.66** |
>
> **Q4: range of ρ_t during training**
>
> **A4:** We conducted a direct comparison of $ρ_t$ statistics (effective gradient magnitude after respective operations) for GradientStabilizer, ZClip, and NormClip under FP16 LLaMA-130M training (lr=3e-3).
>
> - **Key finding: GS is fundamentally distinct from clipping.** The $ρ_t$ distributions differ *qualitatively*, not just quantitatively. Besides,  GS+AdamW leads to a well-behaved raw gradient norm curve ([here](https://drive.google.com/file/d/1VEzmtiiJRkQDFhS3Ee8fbhbQRjz4vC7v/view?usp=sharing)) and smoothly decreasing evaluation loss ([here](https://drive.google.com/file/d/1laiSkp6yU7OlLqTUBv1ZJGX50iPzdszq/view?usp=sharing)) (**In constrast**, AdamW: diverges and clipping methods: decreasing slowly), indicating GS yields a non-degenerate and meaningful gradient magnitude while smoothing away extreme spikes (AdamW max: 10981.17 → GS max: 20.03).
>
> - The full $\rho_t$ trajectory available [here](https://drive.google.com/file/d/1LTCmGj5r_b3_HC_RDt1D39wDINa81hck/view?usp=sharing).
>
> **Table: $\rho_t$ Statistics**
> | Method |P10|P50|P90|Max|
> |---|---|---|---|---|
> | AdamW | 0.29|17.38| 69.64| 10981.17 |
> | AdamW+ZClip |0.27| 0.99|1.0|1.01 |
> | AdamW+NormClip | 0.31 | 0.99 | 1.0 | 1.10 |
> | AdamW+GS | 8.07 | 9.89 | 10.36 | 20.03 |

---

> > ### Author Rebuttal · Reviewer_1jTr · 2026-04-03
> >
> > Thanks for the detailed rebuttal. This significantly improves the paper.
> >
> > The additional comparisons against tuned NormClip and ZClip are helpful and strengthen the empirical case, especially given the sensitivity of clipping to threshold choice. The new mechanism-isolation ablations are also valuable and address one of my main concerns: they make it clearer that the specific combination used in GS is doing more than generic normalization or simple magnitude smoothing. I also appreciate the added diagnostics in RL, where the spike suppression and performance gains are quite compelling.
> >
> > That said, some aspects remain less clear. The comparison to strong baselines is still somewhat limited in scope (e.g., not all domains are covered, and AGC is not included), and the robustness story outside LLMs remains uneven. In particular, the ImageNet results still do not fully support the spike-driven motivation, since catastrophic spikes do not seem to occur there. I also think some of the diagnostic evidence could be more complete (e.g., across seeds or with more detailed trajectories), although I understand this is hard to fully address within the rebuttal.
> >
> > Overall, the rebuttal resolves a substantial part of my concerns and improves my confidence in the method, even if some questions would require more extensive revisions. I will increase my score to 4 (borderline accept).

---

> > > ### Author Response · Authors · 2026-04-03
> > >
> > > Thank you for the detailed and constructive feedback, and for raising the score. We are very glad that the added comparisons and mechanism-isolation ablations helped clarify the empirical strength and distinctiveness of **GradientStabilizer**. We would like to briefly address the remaining points:
> > >
> > > (1) **Regarding ImageNet and the spike-driven motivation.**
> > > - We appreciate this observation. We agree that the spike-driven framing is most directly applicable to LLM and RL training. On ImageNet, the benefit of **GradientStabilizer** is better understood through the variance-dampening mechanism (Lemma 3.2 in Submission): the stabilized magnitude $\rho_t$ continuously contracts gradient magnitude in proportion to the coefficient of variation of gradient norms, regardless of whether catastrophic spikes occur.
> > >
> > > - The benefits of **GradientStabilizer** on ImageNet are further supported by Table 3 (In Submission), where all clipping baselines suffer severe degradation under increased weight decay with Adam (e.g., NormClip, ZClip, and AGC drop to 20.6%, 30.5%, and 0.1%, respectively, on ViT-B at $\mathrm{WD}=5\times10^{-4}$), while **GradientStabilizer** maintains 72.4% accuracy. We will revise the description to present this dual interpretation more explicitly across domains, and we will add gradient-norm trajectory analyses for ImageNet to further characterize the mechanism.
> > >
> > > (2) **Regarding AGC and trajectory-level diagnostics:** We will include a  independently tuned AGC comparisons and more detailed gradient norm trajectory analyses in the revision to provide a more complete empirical evidence.
> > >
> > > Thank you again for the careful and encouraging evaluation. Your feedback has been invaluable in strengthening the paper.
> > >
> > > Best regards,
> > >
> > > The Authors

---

### Official Review · Reviewer_9kJi · 2026-03-08

**Soundness:** 3
**Presentation:** 3
**Significance:** 3
**Originality:** 3
**Overall Recommendation:** 5
**Confidence:** 3

**Summary:**

The paper introduces GradientStabilizer that stabilizes gradients during training. The key idea is to maintain moving averages of the gradient norm $m_t^R$ and its square $v_t^R$, and use its ratio $\rho_t = m_t^R / \sqrt{v_t^R}$ as an estimate for the gradient norm at the current step. The gradients at step t are approximated as $\tilde{g}_t = d_t \rho_t$, where $d_t$ is the direction of the gradient.

The authors perform theoretical analyses of the GradientStabilizer under a fixed mean-variance and a spike model to characterize its behavior. Furthermore, they perform extensive empirical analysis across diverse experiments (language pre-training, image classification, reinforcement learning, time series analysis) in both FP16 and FP4 settings to demonstrate that it performs better than gradient clipping baselines.

**Compliance With Llm Reviewing Policy:**

Affirmed.

**Final Justification:**

The authors resolved my concerns. I have raised my score to accept.

**Key Questions For Authors:**

- What happens when $\kappa$ is too small? Lemma 3.4 would suggest that $\rho_t$ is unbounded. Did the authors check in their experiments?
- Weight decay is typically known to improve performance. However, Table 3 shows that a small weight decay can drastically degrade performance for vanilla Adam. Do the authors understand what is going on here?

**Limitations:**

- Introduces two new hyperparameters: $\gamma_1, \gamma_2$ that control the moving average of gradient norm and its square.
- Unclear if sudden small gradient norms will create a problem.

**Strengths And Weaknesses:**

Strengths:
- The problem, analysis, and experiments are well motivated.
- Theoretical analysis in controlled settings to characterize the behavior of GradientStabilizer
- Extensive experimental analysis across training regimes: language pre-training, image classification, time series forecasting, reinforcement learning in both FP16 and FP4 settings.
- Hyperparameter analysis

Weaknesses:
- Introduces two additional hyperparameters
- Unclear what happens to GradientStabilizer when suddenly a small gradient norm is observed $\kappa << 1$. According to the theoretical results, $\rho_t$ would be unbounded.

---

> ### Author Rebuttal · Authors · 2026-03-31
>
> We sincerely appreciate your positive scores. We provide point-wise responses to your concerns below.
>
> **Q1: "What happens when $\kappa$ is too small? Lemma 3.4 would suggest that $\rho_t$ is unbounded.**
>
> **A1:** Thank you for your comments.
> - We clarify that **Lemma 3.4 is a conditional result that applies specifically to spike steps** (where $R_t ≥ κ·m_{t-1}^{(R)}$ with κ ≫ 1). It is not intended to cover the case of small gradients, and indeed the bound becomes loose when κ is small. However, **Lemma 3.6 provides the unconditional, time-uniform bound** that covers *all* steps — spikes, normal steps, and small-gradient steps alike: $$|\tilde{g}_t|_2 \leq \rho_t \leq \bar{\rho} := \frac{1-\gamma_1}{\sqrt{1-\gamma_2}} \cdot \frac{1}{\sqrt{1 - \gamma_1^2/\gamma_2}}, \quad \forall\, t \geq 1.$$
>
> - This bound depends **only on γ₁ and γ₂** and holds regardless of the instantaneous gradient norm — whether it is extremely large or extremely small.
>
> - **Intuitively**, when a suddenly small gradient appears, the numerator $m_t^{(R)} = γ₁·m_{t-1}^{(R)} + (1−γ₁)·R_t$ is dominated by the historical EMA $m_{t-1}^{(R)}$ (since γ₁ = 0.6 provides substantial momentum), while the denominator $√v_t^{(R)}$ is similarly anchored by historical second moments.
>
> - In the extreme case where $R_t$ = 0, we have: $\rho_t = \frac{\gamma_1}{\sqrt{\gamma_2}} \rho_{t-1}.$
>
> Therefore, $ρ_t$ transitions smoothly rather than blowing up or vanishing. The EMA mechanism inherently prevents abrupt reactions to a single anomalous step.
>
> **Q2:Weight decay is typically known to improve performance. However, Table 3 shows that a small weight decay can drastically degrade performance for vanilla Adam. Do the authors understand what is going on here?**
>
> **A2:** Thank you for your precise observation. The interaction between Adam and weight decay is subtle and has been discussed in the literature (Loshchilov & Hutter, 2017[1]). The key issue is as follows:
>
> - In Adam , weight decay is implemented as an $ℓ₂$ penalty added to the loss, so the gradient becomes $g_t + λθ_t$. Adam's adaptive denominator $√v_t$ then **rescales this combined signal per coordinate**. When λ is non-trivial (e.g., 10⁻⁴ or 5×10⁻⁴), the regularization term $λθ_t$ can dominate the task gradient for certain parameters, distorting the effective learning rate and causing the optimizer to prioritize shrinking weights rather than minimizing the task loss. This effect is amplified in transformer architectures such as ViT-S, where parameter scales vary significantly across layers.
>
> - **Why does GradientStabilizer help?** One plausible mechanism is that, in vanilla Adam, when the task gradient becomes too small or unstable, the coupled decay term $λθ_t$ can become relatively over-dominant. GradientStabilizer mitigates this by stabilizing the gradient magnitude before the Adam update, thereby preserving a healthier balance between the task-gradient signal and the regularization term. More importantly, it also prevents spike-driven corruption of Adam's moment estimates, making the effective adaptive scaling and per-coordinate updates much better conditioned overall.
>
> - Notably, all clipping baselines also fail to address this issue (and some exacerbate it, e.g., AGC drops to 0.1%), because they act only on extreme gradient events rather than structurally stabilizing the update magnitude at every step.
>
> [1] Loshchilov, Ilya, and Frank Hutter. "Decoupled weight decay regularization." arXiv preprint arXiv:1711.05101 (2017).

---

> > ### Author Rebuttal · Reviewer_9kJi · 2026-04-02
> >
> > Thanks for the rebuttal. My concerns are resolved. I am currently going through other reviews and comments and will update my score before the rebuttal period ends.

---

> > > ### Author Response · Authors · 2026-04-02
> > >
> > > We are very glad that our rebuttal has adequately addressed your concerns. We appreciate your time in carefully reviewing both our response and the broader discussion.
> > >
> > > Best regards,
> > >
> > > The authors

---

### Official Review · Reviewer_iJKV · 2026-03-08

**Soundness:** 3
**Presentation:** 3
**Significance:** 3
**Originality:** 3
**Overall Recommendation:** 4
**Confidence:** 4

**Summary:**

GradientStabilizer is a drop-in gradient transformation designed to address training instability caused by gradient-norm spikes. Rather than clipping the gradient when its norm exceeds a threshold (as in classic gradient clipping), the method replaces the instantaneous gradient norm with a statistically stabilized estimate derived from EMA (exponential moving average) statistics of the running gradient norm history. Concretely, it computes the unit direction of the gradient, then scales it by ρt = mR_t / sqrt(vR_t), where mR and vR are EMA estimates of the first and second moments of the gradient norm. This magnitude is provably bounded regardless of spike size, and the paper shows this boundedness propagates into bounded Adam/AMSGrad moment states. Experiments span LLM pre-training (FP16 and FP4), ImageNet classification, reinforcement learning, and time-series forecasting, uniformly showing improvement over clipping baselines.

**Compliance With Llm Reviewing Policy:**

Affirmed.

**Key Questions For Authors:**

see weakness

**Limitations:**

yes

**Strengths And Weaknesses:**

### Strengths

**1. Elegant, well-motivated core idea.**
The decoupling of gradient direction from gradient magnitude is clean and principled. The central observation — that the gradient direction is often reliable while the norm can be highly volatile — is well-supported by the literature and by the paper's own empirical stability analyses (Figure 3, Figure 7). The method is extremely simple to implement (Algorithm 1 is about 5 lines) and adds negligible computational overhead.

**2. Strong and coherent theoretical analysis.**
The paper provides a solid theoretical framework with results that build logically on one another:

* Lemma 3.2 (Variance Dampening) gives a clean population-level interpretation: ρ★ = 1/sqrt(1+c²_v), which decreases as gradient-norm variance increases — a satisfying explanation of why the method works in stationary settings.
* Lemma 3.4 gives a spike-invariant upper bound that decays as 1/κ when spikes are large, meaning the method becomes *more* conservative exactly when the raw gradient is most dangerous.
* Lemma 3.6 gives a time-uniform bound ρ̄ depending only on the EMA hyperparameters (γ1, γ2), entirely independent of the raw gradient sequence. This is a strong unconditional guarantee.
* Theorem 3.9 propagates this into bounded Adam/AMSGrad moment states, and Corollary 3.11 establishes bounded per-step SGD parameter changes.

The proofs are complete, clean, and non-trivial (particularly the Cauchy-Schwarz argument in Lemma 3.6). The heatmaps in Figure 1 provide good visual intuition for how the bound behaves across hyperparameter configurations.

**3. Impressive weight-decay stability result (Table 3).**
This is arguably the most striking empirical finding. All clipping baselines (including ZClip) dramatically degrade ViT-B accuracy as weight decay increases from 0 to 5e-4, with some (AGC, NORM CLIP) collapsing to near-zero accuracy. GradientStabilizer maintains performance (77.3 → 78.7 → 72.4) throughout. The results for ConvNeXt-T and ResNet-50 in Table 9 are equally compelling. This is a qualitatively different kind of stability benefit that has not been highlighted in prior work, and it is likely to be practically important.

**4. Broad empirical coverage and consistent results.**
The experiments are genuinely diverse: LLM pre-training at two scales (130M, 350M), two precision regimes (FP16, FP4), three ImageNet architectures (ViT-B, ConvNeXt-T, ResNet-50), four RL environments (HalfCheetah, Ant, Hopper, Walker2D), and time-series forecasting. GradientStabilizer consistently achieves best or second-best results across all settings. Importantly, ZClip (a strong recent baseline) does not consistently win outside LLM tasks, which highlights a genuine gap that GradientStabilizer fills.

**5. Hyperparameter robustness.**
Figure 6 shows that the maximum validation loss variation across a wide sweep of γ1 ∈ [0.5, 0.8] and γ2 ∈ [0.95, 0.99999] is less than 0.02 — this is a very tight range, suggesting the method is practically robust to its own hyperparameter choices. This is a significant advantage over clipping methods which require careful threshold selection.

**6. Optimizer-agnostic design demonstrated empirically.**
Table 4 shows consistent gains when applying GradientStabilizer to Lion and Adam-Mini, two optimizers with quite different update rules from Adam/AdamW. This meaningfully supports the paper's claim of broad applicability.

---

### Weaknesses

**1. The theoretical guarantees are stability conditions, not convergence results.**
The paper correctly and explicitly acknowledges (Remark 3.10) that it does not claim convergence — only that moment states are bounded, which is a prerequisite typically *assumed* in convergence analyses. However, this means the paper provides no rate guarantees for GradientStabilizer. The question of whether stabilizing the magnitude in this way introduces bias into the descent direction — and at what cost to convergence speed — is left completely open. A non-asymptotic convergence bound, even under simplified assumptions, would substantially strengthen the theoretical contribution.

**2. The method fundamentally changes the effective learning rate.**
GradientStabilizer replaces the gradient magnitude with ρt, which is always ≤ 1 by Lemma 3.2. This is structurally equivalent to applying a time-varying, state-dependent learning rate scaling. The paper does not discuss whether this effective learning rate rescaling needs to be compensated for (e.g., by increasing the nominal learning rate), or whether the empirical gains could be partly attributed to this implicit learning rate effect rather than purely to spike suppression. A controlled experiment varying the nominal learning rate for baselines to match the effective step size of GradientStabilizer would help isolate the mechanism.

**3. Comparison to normalized gradient descent (NGD) is absent.**
The paper notes in Remark 3.3 that in the low-variance regime (cv → 0), GradientStabilizer recovers Normalized Gradient Descent (NGD), which simply uses unit-norm gradients. In the high-variance regime, it uses a reduced magnitude. NGD is a well-known method and a natural baseline — yet it is not included in any experiment. Without this comparison it is unclear how much of the benefit comes from the specific EMA-based magnitude vs. simply normalizing the gradient.

**4. LLM experiments use small-scale models only.**
The LLaMA experiments are at 130M and 350M parameters. These are useful proof-of-concept models but are far below the scale at which gradient spikes are reported to be most damaging (Chowdhery et al. note issues beyond 60B parameters). The paper's claims about benefit for large-scale training are not directly supported. At minimum, a 1B-scale experiment (which is feasible with recent hardware and codebases) would strengthen the case.

**5. ImageNet training is truncated (120 epochs vs. standard 300 epochs).**
The paper notes ImageNet models are trained for 120 epochs "due to the limitation of computing resources." Standard ViT-B and ConvNeXt-T benchmarks use 300 epochs, and the absolute accuracy numbers (79.3% for AdamW+ViT-B vs. the standard ~81.8%) confirm this gap. The gains of GradientStabilizer at 120 epochs may not reflect behavior at convergence, where gradient spikes are likely less frequent. This is a meaningful limitation given ImageNet is a central benchmark.

**6. No ablation on the spike frequency / severity regime.**
The theoretical analysis carefully distinguishes spike events (Rt ≥ κ m^R_{t-1}) from stationary regimes, and the spike-dampening bound explicitly depends on κ. However, no experiment measures how many spike events actually occur during training, how large κ is in practice, or how the gains of GradientStabilizer vary as a function of spike frequency. This connection between the theoretical motivation and the empirical setting is not closed.

**7. The EMA initialization bias is not addressed.**
With m^R_0 = v^R_0 = 0, the stabilized magnitude ρt in the first iterations is computed from heavily biased moment estimates. Adam applies bias correction to address exactly this issue, but GradientStabilizer does not. Early in training, ρt could behave quite differently from its stationary-regime target. The paper does not discuss whether bias correction is needed, or whether the "late warmup" behavior observed in training (the "late wake-up" phenomenon in the training curves) is partly a consequence of this initialization.

**8. Minor clarity issues.**

* Section headings "Theorectical Justification" (p.2) and "Comprison" (pp. 15–16) contain typos.
* Figure 2 is referenced before Figure 3 in the text but appears to show RL results; the caption says HalfCheetah-v4 but Figure 9 also shows this. This is confusing.
* There is no notation table despite the paper introducing a moderately large set of symbols (Rt, m^R_t, v^R_t, ρt, κ, cv, etc.).

---

> ### Author Rebuttal · Authors · 2026-03-31
>
> **GS denotes GradientStabilizer in this rebuttal**
>
> **Q1:Lack convergence results**
>
> **A1:** We agree Section 3 proves stability, not convergence. We add the following non-asymptotic result and will include it in the revised version.
>
> Consider the full-gradient update $x_{t+1} = x_t - \eta_t \rho_t \frac{\nabla f(x_t)}{\lVert\nabla f(x_t)\rVert_2}$ (set to $0$ when $\nabla f(x_t)=0$). Assuming $f$ is $L$-smooth and lower bounded by $f^\star$, by descent lemma: $f(x_{t+1}) \le f(x_t) - \eta_t \rho_t \lVert\nabla f(x_t)\rVert_2 + \frac{L}{2} \eta_t^2 \rho_t^2.$
>
> Summing over $t = 0, \dots, T$,  letting $\Delta := f(x_0) - f^\star$ and Sampling $\tau$ with $\mathbb{P}(\tau = t) = \frac{\eta_t \rho_t}{\sum_{s} \eta_s \rho_s}$,
>
> Then it gives $\mathbb{E} \lVert\nabla f(x_\tau)\rVert_2 \le \frac{\Delta + \frac{L}{2} \sum_{t=0}^{T} \eta_t^2 \rho_t^2}{\sum_{t=0}^{T} \eta_t \rho_t}.$
>
> *Note that* in the full-gradient setting, GS introduces *no directional bias*: it preserves the descent direction exactly and only rescales step length.
>
> Under a non-collapse condition $0 < \rho_{\min} \le \rho_t $ and $\rho_t \le \bar{\rho}$ (our Lemma 3.6)  with constant $\eta_t \equiv \eta$, choosing $\eta = \sqrt{\frac{2\Delta}{L \bar{\rho}^2 (T+1)}}$ yields $\frac{1}{T+1} \sum_{t=0}^{T} \lVert\nabla f(x_t)\rVert_2 \le \frac{\bar{\rho}}{\rho_{\min}} \sqrt{\frac{2L\Delta}{T+1}}.$
>
> Thus the method retains the standard $O(1/\sqrt{T})$ stationarity rate, up to the factor $\bar{\rho} / \rho_{\min}$.
>
> **Q2: Implicit learning rate effect**
>
> **A2:** We thank the reviewer for this important point.
> - We agree that GS changes the update magnitude, and for SGD this can be interpreted as a time-varying scalar $ηρ_t$. However, GS is **not** equivalent to simply using a smaller learning rate. The ratio $ρ_t$ is **state-dependent**: it primarily contracts updates during spike events while leaving non-spiky iterations largely unaffected. A uniformly smaller learning rate suppresses all updates indiscriminately and therefore cannot isolate spike damping from ordinary optimization progress.
>
> - Empirically, the table below shows that tuning the learning rate for AdamW still cannot match GS's performance, confirming that its gains do not arise merely from reducing the effective learning rate.
>     | | lr=5e-4|1e-3|2e-3|
>     |---|--|--|--|
>     |AdamW|32.6|28.7|34.8|
>     |+ GS | -|26.6|- |
>     *(FP4 Training, PPL ↓).*
>
> **Q3: Lack NGD baseline..**
>
> **A3:**  We agree that NGD is a natural baseline. To address this, we have added NGD comparisons on FP16 LLM training (LLaMA), as well as on ImageNet based on AdamW.
>
> - GS consistently outperforms NGD across all settings, confirms that the benefit does not come merely from normalizing the gradient to unit norm.
>
> | |130M|350M|
> |---|--|--|
> |**Ours**|23.14|17.80|
> |**NGD**|24.13|18.58|
>
> | | ViT-B |ConvNeXt-T|
> |---|--|--|
> |**Ours**|79.6|80.1|
> |**NGD**|68.1|76.3|
>
> **Q4: Small-scale LLM models only**
>
> **A4:** We have added LLaMA-1B experiments using 6.6B training tokens, comparing against strong clipping baselines including NormClip and ZClip.
>
> - The results below show that GS outperforms strong baselines by a clear margin at the 1B scale, confirming that its advantage is not limited to small models.
>
> **FP16 Training**
> | Method| Perplexity|
> |---|---|
> |**AdamW** |17.66|
> |**+ ours**|16.06|
> |**+ZClip**|16.82|
> |**+NormClip**|16.89|
>
> **Q5: 300 epochs Training**
>
> **A5:**  We include 300-epoch results on ImageNet below. The additional results confirm that the advantage of GS persists under the standard longer training regime.
> | Method | ViT-B | ConvNeXt |
> |---|---|---|
> |**AdamW**|80.8| 81.8|
> |**+Ours** |81.3| 82.2|
> |**+NormClip** |81.0|81.9|
> |**+ZClip**|80.9|81.9|
>
> **Q6: No Ablation on Spike Frequency/Severity**
>
> **A6:** We agree that explicitly connecting spike frequency/severity to empirical gains would strengthen the paper. However, directly controlling spike frequency and severity is inherently difficult, as spikes emerge from complex interactions between data, model, and optimizer dynamics.
>
> - Alternatively, we provide controlled evidence roughly isolating spike *severity* and *frequency*. Using the same experimental setting in Figure 5 (In Submission), we inject perturbations with increasing magnitude. As anomaly severity increases (1→3), the performance gain of GS consistently increases (0.008→0.025). When varying anomaly **frequency** (5%→20%) yields marginal changes in performance gains.
> |Severity|1|2|3|
> |---|--|--|--|
> |Gain|0.008|0.018|0.025|
> |**Freq**|5%|10%|20%|
> |Gain |0.018|0.0187|0.0188|
>
> **Q7: EMA initialization bias.**
>
> **A7:** We clarify that our **implementation does apply bias correction** to the scalar moment estimates, analogous to Adam. This detail was unfortunately omitted from the manuscript for the convenience of theoretical analysis, and we will revise the method description and pseudocode to state it explicitly.
>
> **Q8:Minor issue**
>
> **A8:** We will correct it in the revised version.

---

> > ### Author Rebuttal · Reviewer_iJKV · 2026-04-02
> >
> > Thank the author for the clarification. All my concerns have been addressed properly. I am willing to increase my score.

---

> > > ### Author Response · Authors · 2026-04-02
> > >
> > > We are very glad that our clarification and additional experiments have fully addressed your concerns. We sincerely appreciate your careful evaluation and are especially grateful for your willingness to raise your score. Your constructive feedback has been very valuable in improving the paper.
> > >
> > > Best regards,
> > >
> > > The authors

---

### Official Review · Reviewer_yuD5 · 2026-03-09

**Soundness:** 2
**Presentation:** 4
**Significance:** 3
**Originality:** 3
**Overall Recommendation:** 4
**Confidence:** 4

**Summary:**

This paper proposes a method called "Gradient Stabilizer", which replaces the original gradient update by multiplying the current gradient direction with a "stabilized magnitude" calculated using the exponential moving average of historical gradient norms. The authors claim that this method outperforms traditional gradient clipping techniques across various tasks, including LLM pre-training, reinforcement learning, and time-series forecasting.

**Compliance With Llm Reviewing Policy:**

Affirmed.

**Final Justification:**

My concerns are mostly addressed by the rebuttal. Given the strengthes listed above, I lean to a score of 4.

**Key Questions For Authors:**

I don't have further questions.

**Limitations:**

See weaknesses.

**Strengths And Weaknesses:**

## Strengths
1. The motivation of the paper is clear, addressing the critical issue of training divergence caused by severe gradient-norm spikes.

2. The proposed algorithm is exceptionally simple, involving only scalar-level EMA computations, making it lightweight and easy to integrate into existing codebases as a drop-in replacement.

3. The empirical evaluation is comprehensive, spanning four major domains: LLMs, vision, reinforcement learning, and time-series forecasting. Notably, in the highly unstable FP4 QAT setting, the proposed method significantly reduces the validation perplexity by approximately 2.5 on the LLaMA-350M model. Furthermore, the experiments demonstrate that the method substantially mitigates Adam's sensitivity to weight-decay strength, which is a highly practical and valuable finding.

## Weaknesses
1. The proposed architecture essentially transplants the standard EMA logic used by the Adam optimizer for coordinate-wise updates (calculating the first and second moments) directly to the global scalar gradient norm. Using a scalar EMA to smooth the gradient magnitude is a marginal algorithmic modification. The authors do not provide an in-depth explanation or theoretical intuition for why this specific formulation is fundamentally better than existing adaptive clipping methods.

2. I disagree with the methodological choice stated in line 231: "For a fair comparison, we do not tune hyperparameters for either the baselines or our method; the same settings are used throughout the paper."  I think a genuinely fair and rigorous empirical assessment requires evaluating each approach at its respective optimal hyperparameter configuration.

3. The authors dedicate the entirety of Section 3 to theoretical proofs, yet they concede, "we do not claim convergence by itself". The primary theoretical claim is merely that the values calculated using the EMA formulas are uniformly bounded. This is mathematically trivial; restricting a scalar with bounded decay rates ranging from 0 to 1  naturally results in an upper bound. Hard clipping provides a strict mathematical upper bound as well. Therefore, the contribution of the theoretical section is unclear. The theoretical results fail to rigorously demonstrate the specific advantages or the necessity of the Gradient Stabilizer over standard clipping mechanisms.

4. The authors do not adequately illustrate exactly how gradient spikes impact LLM pre-training in practice. It would be highly beneficial to see specific, isolated cases where the base AdamW optimizer fails catastrophically due to spikes, but successfully recovers when standard clipping methods are applied. A more granular visualization of the training dynamics during a spike event would significantly strengthen the motivation.

Minors:
The beginning of section 2 is a little bit werid. It looks like there is one missing paragraph at the front.

---

> ### Author Rebuttal · Authors · 2026-03-31
>
> **Q1:The method appears to be a minor extension of Adam’s EMA, .....**
>
> **A1:**We agree that the novelty is *not* EMA by itself. The key point is what the EMA is estimating and how it is used. Adam uses coordinate-wise $m_t/\sqrt{v_t}$ as a preconditioner, which changes the update direction. GradientStabilizer first decomposes the gradient as $g_t = R_t d_t$ with $R_t=\|g_t\|_2$ and $\|d_t\|_2=1$, then keeps $d_t$ and uses $\rho_t=\frac{m_t^{(R)}}{\sqrt{v_t^{(R)}}}$ only as a scalar reliability coefficient for the *radial* part. So the method is not "Adam on a scalar"; it is a direction-preserving radial stabilization rule.
>
> - **Why this particular ratio.** Under the stationary model of Lemma 3.2,  $\rho^\star=\frac{\mathbb{E}[R]}{\sqrt{\mathbb{E}[R^2]}}=\frac{1}{\sqrt{1+c_v^2}}.$ Hence the quantity is scale-free: multiplying all gradient norms by a constant leaves $\rho^\star$ unchanged. It reacts to *relative variability*, not absolute size. Large but stable norms are therefore not additionally penalized because of scale alone, while volatile or spiky norms are automatically damped.
>
> - **Why this differs from adaptive clipping.** The adaptive clipping baselines considered here are still threshold-triggered piecewise rules: AGC clips when a gradient-to-parameter ratio exceeds $\lambda$, and ZClip adjusts only when a z-score exceeds $z_{\mathrm{thres}}$. GradientStabilizer instead uses the statistics to set the update magnitude at *every* step, with no trigger threshold. In short, clipping adapts a threshold, whereas GradientStabilizer adapts the radial magnitude itself. This is also why Lemma 3.4 yields a spike-invariant ceiling, with only a vanishing $1/\kappa$ correction on spike steps. We will revise the paper to make this intuition explicit.
>
> **Q2: A genuinely fair comparison requires ... optimal hyperparameters**
>
> **A2:**  We agree per-method tuning is a valid protocol, but chose the fixed-hyperparameter setting deliberately for two reasons.
>
> - **Practical relevance.** Per-method sweeps are prohibitively expensive at scale. A method that works well with a single fixed configuration across LLM pre-training, vision, RL, and time-series forecasting offers substantially greater practical value.
> - **Fairness.** All baselines use their published default settings; GradientStabilizer's hyperparameters were likewise fixed across all tasks with no per-domain tuning.
>
> To further address your concern, we tuned threshold values for NormClip (most popular) and ZClip (most recent) on FP16 LLM Training and ViT-B based on AdamW. As shown in below Table, **even with per-task optimal tuning, neither baseline surpasses GradientStabilizer's fixed-hyperparameter performance**, further confirming GradientStabilizer's superiority.
>
> | Method |Threshold|FP16 LLM|ViT-B ImageNet|
> |:--|:--:|:--:|:--:|
> |**Ours**|—|**23.32**|**79.6**|
> |NormClip|0.5| 24.23 |79.41|
> |NormClip|1.0| 23.86 |79.50|
> |NormClip|1.5| 24.01 |79.40|
> |NormClip| 2.0 | 25.20 |79.35|
> |ZClip| 1.5 / 1.0 | 24.62 |79.48|
> |ZClip| 2.5 / 1.5 | 23.89 |79.54|
> |ZClip| 3.0 / 2.5 | 23.82 |79.40|
> |ZClip| 4.0 / 3.0 | 24.63 |79.38 |
>
> **Q3: The theoretical results are trivial...**
>
> **A3:** We respectfully disagree that the theoretical analysis is trivial.
>
> -  **The key object is the ratio $\rho_t = m_t^R / \sqrt{v_t^R}$, not a raw EMA.** Lemma 3.2 shows $\rho^\star$ is **scale-free**: multiplying all gradient norms by a constant leaves it unchanged. GradientStabilizer reacts to *relative variability*, not absolute size, which is fundamentally different from threshold-based clipping.
>
> -  **The spike analysis is severity-aware, which threshold-triggered clipping method are not.** Standard clipping such as Normclip and AGC apply a fixed action once a threshold is exceeded—a $10\times$ and $1000\times$ spike both yield the same clipped output. Lemma 3.4 shows the bound of $\rho_t$ at *spike step*  converges to a fixed floor, but increasingly severe spikes receive progressively tighter suppression toward that floor. This $\kappa$-dependent correction has no counterpart in standard clipping.
>
> Therefore, the contribution of section 3 goes beyond boundedness. It identifies a distinct stabilization mechanism:  GradientStabilizer is variance-sensitive in stationary regimes and severity-aware on spike steps.
>
> **Q4: The paper lacks granular visualization ...**
>
> **A4:** Thank you for your suggestion. We show the fine-grained visualizations below. (will include them in the revised version.)
>
> - **Figure A [(Link)](https://drive.google.com/file/d/1Vmf9EO5eAKW-H0QbZc_ktjbjb9poN1mn/view?usp=sharing)** shows frequent large-magnitude spikes under AdamW leading to training divergence.
> - **Figure B [(Link)](https://drive.google.com/file/d/12zSIW8nAZfdPnlImI8u5nvFpGn6jOjXR/view?usp=sharing)** isolates a representative spike event, demonstrating that an abrupt increase in gradient norm directly disrupts the smooth loss decrease.
> - **In contrast**, GradientStabilizer effectively suppresses these spikes.

---

> > ### Author Rebuttal · Reviewer_yuD5 · 2026-04-01
> >
> > Thanks for the rebuttal. I raised my score to 4.

---

> > > ### Author Response · Authors · 2026-04-01
> > >
> > > We are glad that our rebuttal has adequately addressed your concerns. We sincerely appreciate your time and thoughtful reconsideration, and we are especially grateful that you raised your score to 4.
> > >
> > > Best regards,
> > >
> > > Authors

---

### Decision · Program_Chairs · 2026-04-30

**Decision:**

Accept (regular)

**Comment:**

This paper studies a method to stabilize the magnitude of the parameter update by replacing it with a statistically stable one using the moment estimates of the gradient norms. The authors theoretically proved its stability properties and empirically tested the training stability and learning rate sensitivity across various tasks. The reviewers agreed upon positive ratings. Overall I recommend accept.